# On Neural Networks as Infinite Tree-Structured Probabilistic Graphical Models

**Boyao Li**[*]
Department of Biostatistics and Bioinformatics
Duke University
boyao.li@duke.edu

**Alexander J. Thomson**[*]
Department of Computer Science
Duke University
alexander.thomson@duke.edu

**Houssam Nassif**
Meta Inc.
houssamn@meta.com

**Matthew M. Engelhard**[†]
Department of Biostatistics and Bioinformatics
Duke University
m.engelhard@duke.edu

**David Page**[†]
Department of Biostatistics and Bioinformatics
Duke University
david.page@duke.edu

## Abstract

Deep neural networks (DNNs) lack the precise semantics and definitive probabilistic interpretation of probabilistic graphical models (PGMs). In this paper, we propose an innovative solution by constructing infinite tree-structured PGMs that correspond exactly to neural networks. Our research reveals that DNNs, during forward propagation, indeed perform approximations of PGM inference that are precise in this alternative PGM structure. Not only does our research complement existing studies that describe neural networks as kernel machines or infinite-sized Gaussian processes, it also elucidates a more direct approximation that DNNs make to exact inference in PGMs. Potential benefits include improved pedagogy and interpretation of DNNs, and algorithms that can merge the strengths of PGMs and DNNs.

## 1 Introduction

Deep neural networks (DNNs), including large language models, offer state-of-the-art performance on many tasks, but they are difficult to interpret due to their complex structure, large number of latent variables, and the presence of nonlinear activation functions [Buhrmester et al., 2021]. To gain a precise statistical interpretation for DNNs, much progress has been made in linking them to probabilistic graphical models (PGMs). Variational autoencoders (VAEs) [Kingma and Welling, 2014] are an early example; more recent examples include probabilistic dependency graphs [Richardson, 2022] as well as work that relates recurrent neural networks (RNNs) with hidden Markov models (HMMs) [Choe et al., 2017] and convolutional neural networks (CNNs) with Gaussian processes (GPs) [Garriga-Alonso et al., 2018]. When such a connection is possible, potential benefits include:

---

[*]Boyao Li and Alexander J. Thomson contributed equally to this work.
[†]Matthew M. Engelhard and David Page jointly supervised this work.

38th Conference on Neural Information Processing Systems (NeurIPS 2024).

- Clear statistical semantics for a trained DNN model, beyond merely providing the conditional distribution over output variables given input variables. Instead, PGMs provide a joint distribution over all variables including the latent variables.

- Ability to import any PGM algorithm into DNNs, such as belief propagation or MCMC, for example to *reverse any* DNN to use output variables as evidence and variables anywhere earlier as query variables.

- Improved calibration, i.e., improved predictions of probabilities at the output nodes by incorporation of PGM algorithms.

In this paper, we establish such a correspondence between DNNs of any structure and PGMs. Given an arbitrary DNN architecture, we first construct an infinite-width tree-structured PGM. We then demonstrate that during training, the DNN executes approximations of precise inference in the PGM during the forward propagation step. We prove our result exactly in the case of sigmoid activations. We indicate how it can be extended to ReLU activations by building on a prior result of Nair and Hinton [2010]. Because the PGM in our result is a Markov network, the construction can extend even further, to all nonnegative activation functions provided that proper normalization is employed. We argue that modified variants of layer normalization and batch normalization could be viewed as approximations to proper MN normalization in this context, although formal analyses of such approximations are left for future work. Finally, in the context of sigmoid activations we empirically evaluate how the second and third benefits listed above follow from the result, as motivated and summarized now in the next paragraph.

A neural network of any architecture with only sigmoid activations satisfies the definition of a Bayesian network over binary variables—it is a directed acyclic graph with a conditional probability distribution at each node, conditional on the values of the node's parents—and thus is sometimes called a Bayesian belief network (BBN). Nevertheless, it can be shown that the standard gradient used in neural network training (whether using cross-entropy or other common error functions) is inconsistent with the BBN semantics, that is, with the probability distribution defined by this BBN. On the other hand, Gibbs sampling is a training method consistent with the BBN semantics, and hence effective for calibration and for reversing any neural network, but it is woefully inefficient for training. Hamiltonian Monte Carlo (HMC) is more efficient than Gibbs and better fits BBN semantics than SGD. We demonstrate empirically that after training a network quickly using SGD, calibration can be improved by fine-tuning using HMC. The specific HMC algorithm employed here follows directly from the theoretical result, being designed to approximate Gibbs-sampling in the theoretical, infinite-width tree structured Markov network. The degree of approximation is controlled by the value of a single hyperparameter that is also defined based on the theoretical result.

The present paper stands apart from many other theoretical analyses of DNNs that view DNNs purely as *function approximators* and prove theorems about the quality of function approximation. Here we instead show that DNNs may be viewed as statistical models, specifically PGMs. This work is also different from the field of *Bayesian neural networks*, where the goal is to seek and model a probability distribution over neural network parameters. In our work, the neural network itself defines a joint probability distribution over its variables (nodes). Our work therefore is synergistic with Bayesian neural networks but more closely related to older work on learning stochastic neural networks via expectation maximization (EM) [Amari, 1995] or approximate EM [Song et al., 2016].

Although the approach is different, our motivation is similar to that of Dutordoir et al. [2021] and Sun et al. [2020] in their work to link DNNs to deep Gaussian processes (GPs) [Damianou and Lawrence, 2013]. By identifying the forward pass of a DNN with the mean of a deep GP layer, they aim to augment DNNs with advantages of GPs, notably the ability to quantify uncertainty over both output and latent nodes. What distinguishes our work from theirs is that we make the DNN-PGM approximation explicit and include *all* sigmoid DNNs, not just unsupervised belief networks or other specific cases.

All code needed to reproduce our experimental results may be found at `https://github.com/engelhard-lab/DNN_TreePGM`.

## 2 Background: Comparison to Bayesian Networks and Markov Networks

Syntactically a Bayesian network (BN) is a directed acyclic graph, like a neural network, whose nodes are random variables. Here we use capital letters to stand for random variables, and following Russell and Norvig [Russell and Norvig, 2020] and others, we take a statement written using such variables to be a claim for all specific settings of those variables. Semantically, a BN represents a full joint probability distribution over its variables as $P(\vec{V}) = \prod_i P(V_i|pa(V_i))$, where $pa(V_i)$ denotes the parents of variable $V_i$. If the conditional probability distributions (CPDs) $P(V_i|pa(V_i))$ are all logistic regression models, we refer to the network as a sigmoid BN.

It is well known that given sigmoid activation and a cross-entropy error, training a single neuron by gradient descent is identical to training a logistic regression model. Hence, a neural network under such conditions can be viewed as a "stacked logistic regression model", and also as a Bayesian network with logistic regression CPDs at the nodes. Technically, the sigmoid BN has a distribution over the input variables (variables without parents), whereas the neural network does not, and all nodes are treated as random variables. These distributions are easily added, and distributions of the input variables can be viewed as represented by the joint sample over them in our training set.

A Markov network (MN) syntactically is an undirected graph with potentials $\phi_i$ on its cliques, where each potential gives the relative probabilities of the various settings for its variables (the variables in the clique). Semantically, it defines the full joint distribution on the variables as $P(\vec{V}) = \frac{1}{Z} \prod_i \phi_i(\vec{V})$ where the partition function $Z$ is defined as $\sum_{\vec{V}} \prod_i \phi_i(\vec{V})$. It is common to use a loglinear form of the same MN, which can be obtained by treating a setting of the variables in a clique as a binary feature $f_i$, and the natural log of the corresponding entry for that setting in the potential for that clique as a weight $w_i$ on that feature; the equivalent definition of the full joint is then $P(\vec{V}) = \frac{1}{Z} e^{\sum_i w_i f_i(\vec{V})}$. For training and prediction at this point the original graph itself is superfluous.

The potentials of an MN may be on subsets of cliques; in that case we simply multiply all potentials on subsets of a clique to derive the potential on the clique itself. If the MN can be expressed entirely as potentials on edges or individual nodes, we call it a "pairwise" MN. An MN whose variables are all binary is a binary MN.

A DNN of any architecture is, like a Bayesian network, a directed acyclic graph. A sigmoid activation can be understood as a logistic model, thus giving a conditional probability distribution for a binary variable given its parents. Thus, there is a natural interpretation of a DNN with sigmoid activations as a Bayesian network (e.g., Bayesian belief network). Note, however, that when the DNN has multiple, stacked hidden nodes, the values calculated for those nodes in the DNN by its forward pass do not match the values of the corresponding hidden nodes in a Bayesian network. Instead, for the remainder of this paper, we adopt the view that the DNN's forward pass might serve as an approximation to an underlying PGM and explore how said approximation can be precisely characterized. As reviewed in Appendix A, this Bayes net in turn is equivalent to (represents the same probability distribution) as a Markov network where every edge of weight $w$ from variable $A$ to variable $B$ has a potential of the following form:

|          | $B$     | $\neg B$ |
|----------|---------|----------|
| $A$      | $e^w$   | 1        |
| $\neg A$ | 1       | 1        |

For space reasons, we assume the reader is already familiar with the Variable Elimination (VE) algorithm for computing the probability distribution over any query variable(s) given evidence (known values) at other variables in the network. This algorithm is identical for Bayes nets and Markov nets. It repeatedly multiplies together all the potentials (in a Bayes net, conditional probability distributions) involving the variable to be eliminated, and then sums that variable out of the resulting table, until only the query variable(s) remain. Normalization of the resulting table yields the final answer. VE is an exact inference algorithm, meaning its answers are exactly correct.

# 3 The Construction of Tree-structured PGMs

Although both a binary pairwise Markov network (MN) and a Bayesian network (BN) share the same sigmoid functional structure as a DNN with sigmoid activations, it can be shown that the DNN does not in general define the same probability for the output variables given the input variables: forward propagation in the DNN is very fast but yields a different result than VE in the MN or BN, which can be much slower because the inference task is NP-complete. Therefore, if we take the distribution $\mathcal{D}$ defined by the BN or MN to be the correct meaning of the DNN, the DNN must be using an approximation $\mathcal{D}'$ to $\mathcal{D}$. Procedurally, the approximation can be shown to be exactly the following: the DNN repeatedly treats the *expectation* of a variable $V$, given the values of $V$'s parents, as if it were the actual *value* of $V$. Thus previously binary variables in the Bayesian network view and binary features in the Markov network view become continuous. This procedural characterization of the approximation of $\mathcal{D}'$ to $\mathcal{D}$ yields exactly the forward pass in the neural network within the space of a similarly structured PGM, yet, on its own, does not yield a precise joint distribution for said PGM. We instead prefer in the PGM literature to characterize approximate distributions such as $\mathcal{D}'$ with an alternative PGM that precisely corresponds to $\mathcal{D}'$; for example, in some variational methods we may remove edges from a PGM to obtain a simpler PGM in which inference is more efficient. Treewidth-1 (tree-structured or forest-structured) PGMs are among the most desirable because in those models, exact inference by VE or other algorithms becomes efficient. We seek to so characterize the DNN approximation here.

This approach aligns somewhat with the idea of the computation tree that has been used to explore the properties of belief propagation by expressing the relevant message passing operations in the form of a tree [Tatikonda and Jordan, 2002, Ihler et al., 2005, Weitz, 2006]. Naturally the design of the tree structured PGM proposed here differs from the computation trees for belief propagation as we instead aim to capture the behavior of the forward pass of the neural network. Nonetheless, both methods share similar general approaches, the construction of a simpler approximate PGM, and aims, to better understand the theoretical behavior of an approximation to a separate original PGM.

To begin, we consider the Bayesian network view of the DNN. Our first step in this construction is to copy the shared parents in the network into separate nodes whose values are not tied. The algorithm for this step is as follows:

a. Consider the observed nodes in the Bayesian network that correspond to the input of the neural network and their outgoing edges.

b. At each node, for each outgoing edge, create a copy of the current node that is only connected to one of the original node's children with that edge. Since these nodes are observed at this step, these copies do all share the same values. The weights on these edges remain the same.

c. Consider then the children of these nodes. Again, for each outgoing edge, make a copy of this node that is only connected to one child with that edge. In this step, for each copied node, we then also copy the entire subgraph formed by all ancestor nodes of the current node. Note that while weights across copies are tied, the values of the copies of any node are not tied. However, since we also copy the subtree of all input and intermediary hidden nodes relevant to a forward pass up to each copy, the probability of any of these copied nodes being true remains the same across copies (ignoring the influence of any information passed back from their children).

d. We repeat this process across each layer until we have separate trees for each output node in the original deep neural network graph.

This process ultimately creates a graph whose undirected structure is a tree or forest. In the directed structure, trees converge at the output nodes. The probability of any copy of a latent node given the observed input (and ignoring any information passed back through a node's descendant) is the same across all the copies, but when sampling, their values may not be.

The preceding step alone is still not sufficient to accurately express the deep neural network as a PGM. Recall that in the probabilistic graphical model view of the approximation made by the DNN's forward pass, the neural network effectively takes a local average, in place of its actual value, from the immediately previous nodes and passes that information only forward. The following additional step in the construction yields this same behavior. This next step of the construction creates $L$ copies of every non-output node in the network (starting at the output and moving backward) while also copying the entire ancestor subtrees of each of these nodes, as was done in step 1. The weight of a

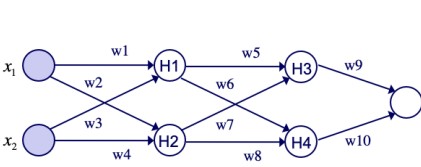
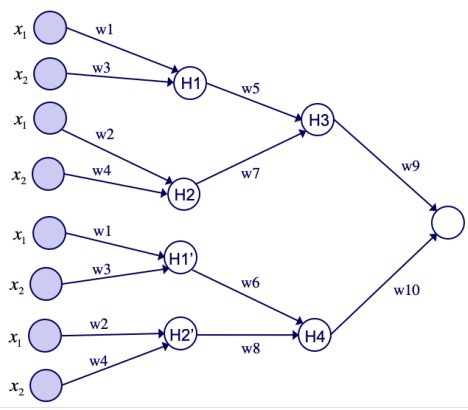

(a) The neural network's graphical structure before applying the first step of this PGM construction.

(b) The neural network's graphical structure after applying the first step of this PGM construction.

Figure 1: The first step of the PGM construction where shared latent parents are separated into copies along with the subtree of their ancestors. Copies of nodes H1 and H2 are made in this example.

copied edges is then set to its original value divided by $L$. Note that this step results in a number of total copies that grows exponentially in the number of layers (i.e. $L$ copies in the 2nd to last layer, $L^2$ copies in the layer before, etc). Detailed algorithms for the two steps in the construction of the infinite tree-structured PGM are presented in Appendix B. As $L$ approaches infinity, we show that both inference and the gradient in this PGM construction matches the forward pass and gradient in the neural network exactly.

This second step in the construction can be thought of intuitively by considering the behavior of sampling in the Bayesian network view. Since we make $L$ copies of each node while also copying the subgraph of its ancestors, these copied nodes all share the same probabilities. As $L$ grows large, even if we sampled every copied node only once, we would expect the average value across these $L$ copies to match the probability of an individual copied node being true. Given that we set the new weights between these copies and their parents as the original weights divided by $L$, the sum of products (new weights times parent values) yields the average parent value multiplied by the original weight. As $L$ goes to infinity, we remove sampling bias and the result exactly matches the value of the sigmoid activation function of the neural network, where this expectation in the PGM view is passed repeatedly to the subsequent neurons. The formal proof of this result, based on variable elimination, is found in Appendix C. There, we show the following:

**Theorem 3.1** (Matching Probabilities). *In the PGM construction, as $L \to \infty$, $P(H = 1|\vec{x}) \to \sigma(\sum_{j=1}^{M} w_j g_j + \sum_{i}^{N} \theta_i \sigma(p_i))$, for an arbitrary latent node $H$ in the DNN that has observed parents $g_1, ..., g_M$ and latent parents $h_1, ..., h_N$ that are true with probabilities $\sigma(p_1), ..., \sigma(p_N)$. Here, $\sigma(\cdot)$ is the logistic sigmoid function and $w_1, ..., w_M$ and $\theta_1, ..., \theta_N$ are the weights on edges between these nodes and $H$.*

The PGM of our construction is a Markov network that always has evidence at the nodes $\vec{X}$ corresponding to the input nodes of the neural network. As such, it is more specifically a conditional random field (CRF). Theorem 1 states the probability that a given node anywhere in the CRF is true given $\vec{X}$ equals the output of that same node in the neural network given input $\vec{X}$. The CRF may also have evidence at the nodes $\vec{Y}$ that correspond to the output nodes of the neural network. Given the form of its potentials, as illustrated at the end of Section 2, the features in the CRF's loglinear form correspond exactly to the edges and are true if and only if the nodes on each end of the edge are true. It follows that the gradient of this CRF can be written as a vector with one entry for each feature $f$ corresponding to each weight $w$ of the neural network, of the form $P(f|\vec{X}) - P(f|\vec{X}, \vec{Y})$. Building on Theorem 1, this gradient of the CRF can be shown to be identical to the gradient of the cross-entropy loss in the neural network: the partial derivative of the cross-entropy loss with respect

to the weight $w$ on an edge, or feature $f$ of the CRF, is $P(f|\vec{X}) - P(f|\vec{X}, \vec{Y})$. This result is more precisely stated below in Theorem 2 below, which is proven in Appendix D.

**Theorem 3.2** (Matching Gradients). *In the PGM construction, as $L \to \infty$, the derivative of the marginal log-likelihood, where all hidden nodes have been summed out, with respect to a given weight exactly matches the derivative of the cross entropy loss in the neural network with respect to the equivalent weight in its structure.*

# 4 Implications and Extensions

We are not claiming that one should actually carry out the PGM construction used in the preceding section, since that PGM is infinite. Rather, its contribution is to give precise semantics to an entire neural network, as a joint probability distribution over all its variables, not merely as a machine computing the probability of its output variables given the input variables. Any Markov network, including any CRF, precisely defines a joint probability distribution over all its variables, hidden or observed, in a standard, well-known fashion. The particular CRF we constructed is the *right* one in the very specific sense that it agrees with the neural network exactly in the gradients both use for training (Theorem 2). While the CRF is infinite, it is built using the original neural network as a template in a straightforward fashion and is tree-structured, and hence it is easy to understand. Beyond these contributions to pedagogy and comprehensibility, are there other applications of the theoretical results?

One application is an ability to use standard PGM algorithms such as Markov chain Monte Carlo (MCMC) to sample latent variables given observed values of input and output variables, such as for producing confidence intervals or understanding relationships among variables. One could already do so using Gibbs sampling in the BN or MN directly represented by the DNN itself (which we will call the "direct PGM"), but then one wouldn't be using the BN or MN with respect to which SGD training in the DNN is correct. For that, our result has shown that one instead needs to use Gibbs sampling in the infinite tree-structured PGM, which is impractical. Nevertheless, for any variable $V$ in the original DNN, on each iteration a Gibbs sampler takes infinitely many samples of $V$ given infinitely many samples of each of the members of $V$'s Markov blanket in the original DNN. By treating the variables of the original DNN as continuous, with their values approximating their sampled probabilities in the Gibbs sampler, we can instead apply Hamiltonian Monte Carlo or other MCMC methods for continuous variables in the much smaller DNN structure. We explore this approach empirically rather than theoretically in the next section. Another, related application of our result is that one could further fine-tune the trained DNN using other PGM algorithms, such as contrastive divergence. We also explore this use in the next section.

One might object that most results in this paper use sigmoid activation functions. Nair and Hinton showed that rectified linear units (ReLU) might be thought of as a combination of infinitely many sigmoid units with varying biases [Nair and Hinton, 2010]. Hence our result in the previous section can be extended to ReLU activations by the same argument. More generally, with any non-negative activation function that can yield values greater than one, while our BN argument no longer holds, the MN version of the argument can be extended. An MN already requires normalization to represent a probability distribution. While Batch Normalization and Layer Normalization typically are motivated procedurally, to keep nodes from "saturating," and consequently to keep gradients from "exploding" or "vanishing," as the names suggest, they might also be used to bring variables into the range $[0, 1]$ and hence to being considered as probabilities. Consider an idealized variant of these that begins by normalizing all the values coming from a node $h$ of a neural network, over a given minibatch, to sum to $1.0$; the argument can be extended to a set of $h$ and all its siblings in a layer (or other portion of the network structure) assumed to share their properties. It is easily shown that if the parents of any node $h$ in the neural network provide to $h$ approximate probabilities that those parent variables are true in the distribution defined by the Markov network given the inputs, then $h$ in turn provides to its children an approximate probability that $h$ is true in the distribution defined by the Markov network given the inputs. Use of a modified Batch or Layer Normalization still would be only approximate and hence adds an additional source of approximation to the result of the preceding section. Detailed consideration of other activation functions is left for further work; in the next section we return to the sigmoid case.

# 5 Application of the Theory: A New Hamiltonian Monte Carlo Algorithm

To illustrate the potential utility of the infinite tree-structured PGM view of a DNN, in this section we pursue one of its implications in greater depth; other implications for further study are summarized in the Conclusion. We have already noted we can view forward propagation in an all-sigmoid DNN as exact inference in a tree-structured PGM, such that the CPD of each hidden variable is a logistic regression. In other words, each hidden node is a Bernoulli random variable, with parameter $\lambda$ being a sigmoid activation (i.e. logistic function) applied to a linear function of the parent nodes. This view suggests alternative learning or fine-tuning algorithms such as contrastive divergence (CD) [Carreira-Perpinan and Hinton, 2005, Bengio and Delalleau, 2009, Sutskever and Tieleman, 2010]. CD in a CRF uses MCMC inference with many MCMC chains to estimate the joint probability over the hidden variables given the evidence (the input and output variables in a standard DNN), and then takes a gradient step based on the results of this inference. But to increase speed, CD-$n$ advances each MCMC chain only $n$ steps before the next gradient step, with CD-1 often being employed. CD has a natural advantage over SGD, which samples the hidden variable values using only evidence in input values; instead, MCMC in CD uses *all* the available evidence, both at input and output variables. Unfortunately, if the MCMC algorithm employed is Gibbs sampling on the many hidden variables found in a typical neural network, then it suffers from high cost in computational resources. MCMC has now advanced far beyond Gibbs sampling with methods such as Hamiltonian Monte Carlo (HMC), but HMC samples values in $[0, 1]$ rather than $\{0, 1\}$. Neal [2012] first applied HMC to neural nets to sample the weights in a Bayesian approach, but still used Gibbs sampling on the hidden variables. Our theoretical results for the first time justify the use of HMC over the hidden variables rather than Gibbs sampling in a DNN, as follows.

Recall that the DNN is itself a BN with sigmoid CPDs, but if we take the values of the hidden variables to be binary then DNN training is not correct with respect to this BN. Instead, based on the correctness of our infinite tree-structured PGM, the probabilistic behavior of one hidden node in the BN is the result of sampling values across its $L$ copies in the PGM. Within any copy, the value of the hidden node follows the Bernoulli distribution with the same probability distribution as the other copies, determined by the parent nodes. Since all the copies share the same parent nodes by the construction and are sampled independently, the sample average follows a normal distribution as the asymptotic distribution when $L \to \infty$ by the central limit theorem. In practice, $L$ is finite and this normal distribution is a reasonable approximation to the distribution of the hidden node. Thus in the BN, whose variables correspond exactly to those of the DNN, the variables have domain [0,1] rather than $\{0, 1\}$, as desired. We next precisely define this BN and the resulting HMC algorithm.

## 5.1 Learning via Contrastive Divergence with Hamiltonian Monte Carlo Sampling

Consider a Bayesian network composed of input variables $\boldsymbol{x} = \boldsymbol{h}_0$, a sequence of layers of hidden variables $\boldsymbol{h}_1, ..., \boldsymbol{h}_K$, and output variables $\boldsymbol{y}$. Each pair of consecutive layers forms a bipartite subgraph of the network as a whole, and the variables $\boldsymbol{h}_i = (h_{i1}, ..., h_{iM_i})$ follow a multivariate normal distribution with parameters $\boldsymbol{p}_i = (p_{i1}, ..., p_{iM_i})$ that depend on variables in the previous layer $\boldsymbol{h}_{i-1}$ as follows:

$$h_{ij} \sim \mathcal{N}(p_{ij}, p_{ij}(1 - p_{ij})/L), \text{ where } \boldsymbol{p_i} = \sigma(\boldsymbol{W}_{i-1}\boldsymbol{h}_{i-1} + \boldsymbol{b}_{i-1}), \tag{1}$$

where $\sigma : \mathbb{R} \to (0, 1)$ is a non-linearity – here the logistic function – that is applied element-wise, and $\boldsymbol{\theta}_i = (\boldsymbol{W}_i, \boldsymbol{b}_i)$ are parameters to be learned. The distribution in equation (1) is motivated by supposing that $h_{ij}$ is the average of $L$ copies of the corresponding node in the PGM, each of which is 1 with probability $p_{ij}$ and zero otherwise, then applying the normal approximation to the binomial distribution. Importantly, this approximation is valid only for large $L$.

For a complete setting of the variables $\{\boldsymbol{x}, \boldsymbol{h}, \boldsymbol{y}\}$, where $\boldsymbol{h} = \{\boldsymbol{h}_1, ..., \boldsymbol{h}_K\}$, and parameters $\boldsymbol{\theta} = \{\boldsymbol{\theta}_i\}_{i=0}^K$, the likelihood $p(\boldsymbol{y}, \boldsymbol{h}|\boldsymbol{x}; \boldsymbol{\theta})$ may be decomposed as:

$$p(\boldsymbol{y}, \boldsymbol{h}|\boldsymbol{x}; \boldsymbol{\theta}) = p(\boldsymbol{y}|\boldsymbol{h}_K; \boldsymbol{\theta}_K) \cdot \prod_{i=1}^{K} \prod_{j=1}^{M_i} p_{\mathcal{N}}(h_{ij}|p_{ij}(\boldsymbol{h}_{i-1}; \boldsymbol{\theta}_{i-1})), \tag{2}$$

where $p_{\mathcal{N}}(\cdot|\cdot)$ denotes the normal density, and a specific form for $p(\boldsymbol{y}|\boldsymbol{h}_K; \boldsymbol{\theta}_K)$ has been omitted to allow variability in the output variables. In our experiments, $\boldsymbol{y}$ is a Bernoulli(binary) or categorical random variable parameterized via the logistic(sigmoid) or softmax function, respectively.

Let $\boldsymbol{h}^{(0)}, \boldsymbol{h}^{(1)}, \boldsymbol{h}^{(2)}, ...$ denote a chain of MCMC samples of the complete setting of hidden variables in the neural network. As previously noted, we allow hidden variables $h_{ij} \in (0, 1)$ for $i \in \{1, ..., K\}$ and $j \in \{1, ..., M_i\}$, and use Hamiltonian Monte Carlo (HMC) to generate the next state due to its fast convergence. Since HMC samples are unbounded, we sample the *logit* associated with $h_{ij} \in (0, 1)$, i.e. $\sigma^{-1}(h_{ij}) \in (-\infty, \infty)$, rather than sampling the $h_{ij}$ directly.

The HMC trajectories are defined by Hamilton's Equations:

$$\frac{d\rho_i}{dt} = \frac{\partial H}{\partial \mu_i} \qquad\qquad \frac{d\mu_i}{dt} = -\frac{\partial H}{\partial \rho_i} \qquad (3)$$

where $\rho_i, \mu_i$ are the $i$th component of the position and momentum vector. They are intermediate variables used to generate a new state for the MCMC chain. The Hamiltonian $H$ is

$$H = H(\boldsymbol{\rho}, \boldsymbol{\mu}) = U(\boldsymbol{\rho}) + \frac{1}{2}\boldsymbol{\mu}^T M^{-1}\boldsymbol{\mu} \qquad (4)$$

where $M^{-1}$ is a positive definite convariance matrix and acts as a metric to rotate and scale the target distribution, which is usually set to identity matrix in practice. Defining the position $\boldsymbol{\rho} = \boldsymbol{h}$, the complete set of hidden variables of the network, we have that the potential energy $U$ is the negative log-likelihood associated with equation (2):

$$U(\boldsymbol{h}) = -\log p(\boldsymbol{y}, \boldsymbol{h}|\boldsymbol{x}; \boldsymbol{\theta}) = -\log p(\boldsymbol{y}|\boldsymbol{h}_K; \boldsymbol{\theta}_K) - \sum_{i=1}^{K}\sum_{j=1}^{M_i} \log p_{\mathcal{N}}(h_{ij}|p_{ij}(\boldsymbol{h}_{i-1}; \boldsymbol{\theta}_{i-1})). \quad (5)$$

We set the leap frog size $l > 0$, step size $\Delta t > 0$. A description of the HMC trajectories (*i.e.*, evolution of $\boldsymbol{h}$) is provided in Appendix E.

The initial state of the chain $\boldsymbol{h}^{(0)}$ is drawn with a simple forward pass through the network, ignoring the output variables; in other words, we have $h_{ij}^{(0)} \sim \mathcal{N}(\sigma(\boldsymbol{W}_{i-1}^{(0)}\boldsymbol{h}_{i-1}^{(0)} + \boldsymbol{b}_{i-1}^{(0)})_j)$ for $i \in \{1, ...K\}$, where $\boldsymbol{h}_0 = \boldsymbol{x}$ are the input variables, and the values of $\boldsymbol{W}_i^{(0)}$ and $\boldsymbol{b}_i^{(0)}$ are manually set or drawn from a standard normal or uniform distribution. We update $\boldsymbol{h}$ through a number of burn-in steps before beginning to update our parameters to ensure that $\boldsymbol{h}$ is first consistent with evidence from the output variables. After $k$ steps, corresponding to CD-$k$, we define the loss based on equation (2):

$$\mathcal{L}(\boldsymbol{\theta}^{(n)}) = -\log p(\boldsymbol{y}, \boldsymbol{h}|\boldsymbol{x}; \boldsymbol{\theta}^{(n)}). \qquad (6)$$

We then apply the following gradients to update the parameters $\{\boldsymbol{W}_i^{(n)}\}_{i=0}^{K}$ and $\{\boldsymbol{b}_i^{(n)}\}_{i=0}^{K}$:

$$\boldsymbol{W}_i^{(n+1)} = \boldsymbol{W}_i^{(n)} - \eta\frac{\partial \mathcal{L}}{\partial \boldsymbol{W}_i^{(n)}} \qquad\qquad \boldsymbol{b}_i^{(n+1)} = \boldsymbol{b}_i^{(n)} - \eta\frac{\partial \mathcal{L}}{\partial \boldsymbol{b}_i^{(n)}} \qquad (7)$$

where $\eta$ is the learning rate. Algorithm 3 (see Appendix F) summarizes this procedure.

## 5.2 Experimental Results

The previous section discussed the modeling of the HMC learning algorithm inspired by the construction of tree-structured PGMs. This section compares the proposed algorithm to Gibbs sampling and SGD training with DNNs in both the synthetic experiments and experiments on Covertype dataset [Blackard, 1998], which is a real-world dataset for classification. They are designed to illustrate how the HMC-based algorithm could fine-tune and improve the calibration of DNNs. The experimental setup and additional experiments are described in Appendix. An internal cluster of GPUs was employed for all experiments, and part of run-times are provided in the appendix; as anticipated, SGD is faster than HMC, which is faster than Gibbs.

### 5.2.1 Synthetic experiments

The synthetic datasets are generated by simple BNs and MNs with their weights in different ranges, which are used to define the conditional probabilistic distributions for BNs and potentials for MNs. Each dataset contains 1000 data points $\{(\boldsymbol{X}_i, y_i)\}, i = 1, 2, ..., 1000$, where each input $\boldsymbol{X}_i \in \{0, 1\}^n$ is a binary vector with $n$ dimension and each output $y_i \in \{0, 1\}$ is a binary value. The true

probabilistic distribution $P(y|\boldsymbol{X})$ of the corresponding BN/MN is calculated by sampling or applying the VE algorithm on it.

To explore how the proposed algorithm performs in model calibration, a DNN is first trained with SGD for 100 or 1000 epochs, and then fine-tuned by Gibbs or HMC with different $L$'s for 20 epochs based on the trained DNN model. Here $L$ defines the normal distribution for hidden nodes in Eqn. 1 and is explored across the set of values: $\{10, 100, 1000\}$. The calibration is assessed by mean absolute error (MAE) in all the synthetic experiments and compared between non-extra fine-tuning (shown in the "DNN" column in Table 1) and fine-tuning with Gibbs or HMC. Since the ground truth of $P(y|\boldsymbol{X})$ in the synthetic dataset can be achieved from the BN/MN, the MAE is calculated by comparing the predicted $P(y|\boldsymbol{X})$ from the finetuned network and the true probability.

Table 1 shows that in general, DNN results tend to get worse with additional training, particularly with smaller weights, and the HMC-based fine-tuning approaches can mitigate this negative impact of additional training on the model calibration. Across all the HMC with different $L$'s, HMC ($L$=10) performs better than the others and DNN training itself for BNs and MNs with smaller weights. Additionally, the MAE of HMC ($L$=10) tends to be similar to Gibbs but runs much faster, especially in the BN simulations, whereas HMC ($L$=1000) is more similar to the NN. This is consistent with what we have argued in the theory that when $L$ goes smaller, the number of the sampled copies for each hidden node decreases in our tree-PGM construction and HMC sampling performs more similar to Gibbs sampling; and as $L$ increases, the probability of each hidden node given the input approaches the result of the DNN forward propagation and thus HMC performs more similar to DNN training.

Table 1: Calibration performance on synthetic datasets. Experiments are run on each dataset 100 times to avoid randomness. T-tests are used to test whether Gibbs and HMC have smaller MAE than SGD, and highlighted cells mean that it is statistically significant to support the hypothesis.

| Data (Weight) | # Train Epochs | Average Mean Absolute Error ($\times 10^{-3}$) (p-value) | | | | |
| --- | --- | --- | --- | --- | --- | --- |
| | | DNN | Gibbs | HMC-10 | HMC-100 | HMC-1000 |
| BN (0.3) | 100 | 6.593 | 16.09 (1.0000) | 5.300 (<0.0001) | 6.864 (0.9982) | 6.658 (1.0000) |
| | 1000 | 34.44 | 36.69 (0.9916) | 23.53 (<0.0001) | 34.96 (1.0000) | 34.55 (1.0000) |
| BN (1) | 100 | 22.90 | 20.84 (0.0011) | 22.48 (<0.0001) | 24.17 (1.0000) | 22.95 (0.9928) |
| | 1000 | 42.59 | 33.64 (<0.0001) | 33.07 (<0.0001) | 43.03 (1.0000) | 42.63 (0.9995) |
| BN (3) | 100 | 72.76 | 76.12 (1.0000) | 76.54 (1.0000) | 72.62 (<0.0001) | 72.63 (<0.0001) |
| | 1000 | 28.28 | 32.59 (1.0000) | 32.98 (1.0000) | 28.84 (1.0000) | 28.40 (1.0000) |
| BN (10) | 100 | 186.0 | 192.8 (1.0000) | 196.1 (1.0000) | 184.6 (<0.0001) | 184.8 (<0.0001) |
| | 1000 | 54.89 | 79.69 (1.0000) | 72.64 (1.0000) | 54.81 (0.1266) | 54.63 (<0.0001) |
| MN (0.3) | 100 | 6.031 | 14.03 (1.0000) | 4.515 (<0.0001) | 6.382 (1.0000) | 6.070 (1.0000) |
| | 1000 | 38.11 | 34.83 (<0.0001) | 26.54 (<0.0001) | 38.71 (1.0000) | 38.22 (1.0000) |
| MN (1) | 100 | 9.671 | 17.81 (1.0000) | 8.887 (<0.0001) | 9.018 (<0.0001) | 9.284 (<0.0001) |
| | 1000 | 27.80 | 32.44 (1.0000) | 19.92 (<0.0001) | 27.98 (0.9994) | 27.73 (<0.0001) |
| MN (3) | 100 | 8.677 | 23.60 (1.0000) | 5.685 (<0.0001) | 5.912 (<0.0001) | 5.964 (<0.0001) |
| | 1000 | 5.413 | 28.03 (1.0000) | 5.792 (0.9957) | 5.671 (1.0000) | 5.443 (1.0000) |

### 5.2.2 Covertype Experiments

Similar experiments are also run on the Covertype dataset to compare the calibration of SGD in DNNs, Gibbs and the HMC-based algorithm. Since the ground truth for the distribution of $P(y|\boldsymbol{X})$ cannot be found, the metric for the calibration used in this experiment is the expected calibration error (ECE), which is a common metric for model calibration. To simplify the classification task, we choose the data with label 1 and 2 and build two binary subsets, each of which contains 1000 data points. Similarly, the number of training epochs is also 100 or 1000, while the fine-tuning epochs shown in Table 2 is 20.

Table 2 shows that HMC with $L = 10$ fine-tuning generally performs better than DNN results, and HMC with $L = 1000$ has the similar ECE as that in DNN. It meets the conclusion made in the synthetic experiments. Gibbs sampling, however, could perform worse than just using DNN. It could be because Gibbs may be too far removed from the DNN, whereas our proposed HMC is more in the middle. This suggests perhaps future work testing the gradual shift from DNN to HMC to Gibbs.

Table 2: Calibration performance on Covertype datasets. Highlighted cells show the best calibrations among each row.

| Data | # Train Epochs | Test Expected Calibration Error ($\times 10^{-2}$) | | | | |
|---|---|---|---|---|---|---|
| | | DNN | Gibbs | HMC-10 | HMC-100 | HMC-1000 |
| Covertype (label 1) | 100 | 4.207 | 4.229 | 2.352 | 3.893 | 3.987 |
| | 1000 | 10.85 | 8.513 | 6.875 | 7.730 | 11.60 |
| Covertype (label 2) | 100 | 4.268 | 7.796 | 7.719 | 4.913 | 4.354 |
| | 1000 | 6.634 | 14.67 | 5.233 | 5.394 | 7.713 |

## 6 Conclusion, Limitations, and Future Work

In this work, we have established a new connection between DNNs and PGMs by constructing an infinite-width tree-structured PGM corresponding to any given DNN architecture, then showing that inference in this PGM corresponds exactly to forward propagation in the DNN given sigmoid activation functions. This theoretical result is valuable in its own right, as it provides new perspective that may help us understand and explain relationships between PGMs and DNNs. Moreover, we anticipate it will inspire new algorithms that merge strengths of PGMs and DNNs. We have explored one such algorithm, a novel HMC-based algorithm for DNN training or fine-tuning motivated by our PGM construction, and we illustrated how it can be used to improve to improve DNN calibration.

Limitations of the present work and directions for future work include establishing formal results about how closely batch- and layer-normalization can be modified to approximate Markov network normalization when using non-sigmoid activations, establishing theoretical results relating HMC in the neural network to Gibbs sampling in the large treewidth-1 Markov network, and obtaining empirical results for HMC with non-sigmoid activations. Also of great interest is comparing HMC and other PGM algorithms to Shapley values, Integrated Gradients, and other approaches for assessing the relationship of some latent variables to each other or to inputs and/or outputs in a neural network. We note that the large treewidth-1 PGM is a substantial approximation to the *direct* PGM of a DNN – in other words, the PGM whose structure exactly matches that of the DNN. In future work, we will explore other DNN fine-tuning methods, perhaps based on loopy belief propagation or other approximate algorithms often used in PGMs, that may allow us to more closely approximate inference in this direct PGM.

Another direction for further work is in the original motivation for this work. Both DNNs and PGMs are often used to model different components of very large systems, such as the entire gene regulatory network in humans. For example, in the National Human Genome Research Institute (NHGRI) program Impact of Genetic Variation on Function (IGVF), different groups are building models of different parts of gene regulation, from genotypic variants or CRISPRi perturbations of the genome, to resulting changes in transcription factor binding or chromatin remodeling, to post-translational modifications, all the way to phenotypes characterized by changes in the expression of genes in other parts of the genome IGVF Consortium [2024]. Some of these component models are DNNs and others are PGMs. As a community we know from years of experience with PGMs that passing the outputs of one model to the inputs of another model is typically less effective than concatenating them into a larger model and fine-tuning and using this resulting model. But this concatenation and fine-tuning and usage could not be done with a mixture of PGM and DNN components until now. Having an understanding of DNN components as PGMs enables their combination with PGM components, and then performing fine-tuning and inference in the larger models using algorithms such as the new HMC algorithm theoretically justified, developed, and then evaluated in this paper. Furthermore, the same HMC approach can be employed to reason just as easily from desired gene expression changes at the output nodes back to variants or perturbations at the input nodes that are predictive of the desired changes. Ordinarily, to reason in reverse in this way in a DNN would require special invertible architectures or training of DNNs that operate only in the other direction such as diffusion. Experiments evaluating all these uses of HMC (or other approximate algorithms in PGMs such as loopy belief propagation or other message passing methods) are left for future work.

## Acknowledgements

The authors would like to thank Sayan Mukherjee, Samuel I. Berchuck, Youngsoo Baek, David B. Dunson, Andrew S. Allen, William H. Majoros, Jude W. Shavlik, Sriraam Natarajan, David E. Carlson, Kouros Owzar, and Juan Restrepo for their helpful discussion about the theoretical work. We are also grateful to Mengyue Han and Jinyi Zhou for their technical support.

This project is in part supported by Impact of Genomic Variation on Function (IGVF) Consortium of the National Institutes of Health via grant U01HG011967.

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

# A  Bayesian Belief Net and Markov Net Equivalence

We don't claim the following theorem is new, but we provide a proof because it captures several components of common knowledge to which we couldn't find a single reference.

**Theorem A.1.** *Let $N$ be a Bayesian belief network whose underlying undirected graph has treewidth 1, and let $w_{AB}$ denote the coefficient of variable $A$ in the logistic CPD for its child $B$. Let $M$ be a binary pairwise Markov random field with the same nodes and edges (now undirected) as $N$. Let $M$'s potentials all have the value $e^{w_{AB}}$ if the nodes $A$ and $B$ on either side of edge $AB$ are true, and the value $1$ otherwise. $M$ and $N$ represent the same joint probability distribution over their nodes.*

*Proof.* According to $M$ the probability of a setting $\vec{V}$ of its variables is

$$\frac{1}{Z}\Pi_i \phi_i(\vec{V})$$

where $\phi_i$ are the potentials in $M$, and $Z$ is the partition function, defined as

$$Z = \Sigma_{\vec{V}}\Pi_i \phi_i(\vec{V})$$

We use $DOM(\phi)$ to designate the variables in a potential $\phi$. Because the nodes and structures of $M$ and $N$ agree, we will refer to the parents, children, ancestors, and descendants of any node in $M$ to designate the corresponding nodes in $N$. Likewise we will refer to the input and output variables of $M$ as those nodes of $N$ that have no parents and no children, respectively. Because $M$ has treewidth 1, each node of $M$ d-separates its set of ancestors from its set of descendants and indeed from all other nodes in $M$. As a result, it is known that the partition function can be computed efficiently in treewidth-1 Markov networks, for example by the following recursive procedure $f$ defined below. Let $V_0$ be the empty set of variables, and let $V_1$ be the input variables of $M$. Let $Ch(V)$ denote the children of any set $V$ of variables in $M$, and similarly let $Pa(V)$ denote the parents of $V$. For convenience, when $V$ is a singleton we drop the set notation and let $V$ denote the variable itself. For all natural numbers $i \geq 0$:

$$f(V_i) = \Pi_{N \in Ch(V_i)}\Sigma_{N=0,1}\Pi_{\phi_j:DOM(\phi_j)\subseteq V_i, DOM(\phi_j)\not\subseteq V_{i-1}}\phi_j(V_i)$$

$$f(V_{m+1}) = 1$$

where $V_m$ is not the full set of variable in $M$ but $V_{m+1}$ is the full set. Then $Z = f(V_1)$.

For each variable $v \in \vec{V}$, we can multiply the potentials on the edges between $v$ and its parents, to get a single potential $\phi_{\{v,Pa(v)\}}$ over $\{v, Pa(v)\}$. For a given setting of the parents of $v$ in $\vec{V}$, let $\phi_{v|Pa(v)}$ denote the result of conditioning on this setting of the parents, and let $\phi_{v,\neg v|Pa(v)}$ denote the result of summing out variable $v$. Using these product potentials of $M$, and given the method above for computing $Z$ for a tree-structured Markov network, we can define the probability of a particular setting $\vec{V}$ as

$$P(\vec{V}) = \Pi_{v \in \vec{V}}\frac{\phi_{v|Pa(v)}}{\phi_{v,\neg v|Pa(v)}}$$

These terms are exactly the terms of the logistic conditional probabilities of the Bayesian belief network $N$:

$$P(\vec{V}) = \Pi_{v \in \vec{V}}P(v|Pa(v))$$

$\square$

Note that in general when converting a Bayes net structure to a Markov net structure, to empower the Markov net to represent any probability distribution representable by the Bayes net we have to moralize. A corollary of the above theorem is that in the special case where the Bayes net uses only sigmoid activations, and its underlying undirected graph is tree-structured, moralization is not required.

# B  Step 1 and Step 2 Construction Algorithms

In the following, calls to add to sets $V$ or $E$ (or to check if either set contains a vertex or edge) immediately edits/checks the respective set object that they reference.

---

**Algorithm 1** Step 1 of the PGM Construction

---

**Require:** A list $H$ of the output vertices of the DNN's DAG $G$
**Ensure:** A set of vertices $V$ and edges $E$ of the DNN's original DAG transformed to a tree structured graph, as shown in Figure 1
1: let $V$ be an empty set of vertices
2: let $E$ be an empty set of edges
3: $V, E \leftarrow$ DNN_TREE($H, G, V, E,$ ' ')
4: **procedure** DNN_TREE($H, G, V, E, child$)       ▷ unroll the DNN graph into a tree(s) layerwise
5:   **for each** $vertex$ **in** $H$ **do**
6:     $new \leftarrow vertex$
7:     **while** $new$ **in** $V$ **do**             ▷ create a new copy of this vertex
8:       $new \leftarrow new +$ '$\prime$' character
9:     **end while**
10:     $V$.add($new$)
11:     **if** $child$ is labelled **then**               ▷ ie child is not ' '
12:       $weight \leftarrow G$.getEdge($vertex, child$.removeAll('$\prime$'))   ▷ get child's original vertex
13:       let $e$ be an edge between $new$ and $child$   ▷ an edge between the parent copy and the child
14:       E.add($e$), E.addWeight($e, weight$)
15:     **end if**
16:     DNN_TREE($G$.parents($vertex$), $G, V, E, new$)    ▷ create unrolled subtrees for each parent
17:   **end for**
18:   **return** V, E
19: **end procedure**

---

**Algorithm 2** Step 2 of the PGM Construction

---

**Require:** A graph $G$ of the vertices and edges created in Step 1, a list $H$ of the DNN's output vertices, an integer $l$
**Ensure:** Vertices $V$ and edges $E$ of the final tree structure graph with $l$ copies of each parent node from Step 1
1: let $V$ be an empty set of vertices
2: let $E$ be an empty set of edges
3: **for each** $vertex$ **in** $H$ **do**
4:   $V', E' \leftarrow$ DNN_COPY($vertex, G, vertex, V, E, l$)
5:   $V \leftarrow V$.union($V'$), $E \leftarrow E$.union($E'$)
6: **end for**
7: **procedure** DNN_COPY($current, G, copy, V, E, l$)
8:   $V$.add($copy$)
9:   **for each** $parent$ **in** $G$.parents($current$) **do**
10:    $weight \leftarrow G$.getWeight($parent, current$)
11:    **for** $i \leftarrow 1$ to $l$ **do**
12:     $n \leftarrow parent +$ '-' $+$ i           ▷ create the ith copy in the current subtree
13:     **while** $n$ in $V$ **do**       ▷ create a unique label so the graph retains its tree-structure
14:       $n \leftarrow n +$ '$\prime$'
15:     **end while**
16:     let $e$ be an edge between $copy$ and $n$   ▷ connect this new copied node to the previous
17:     E.add($e$), E.addWeight($e, weight/l$)
18:     DNN_COPY($parent, G, n, V, E, l$)     ▷ create copies in the remaining subtrees
19:    **end for**
20:   **end for**
21:   **return** $V, E$
22: **end procedure**

---

# C  A Proof Using Variable Elimination

In order to prove that as $L$ goes to infinity, this PGM construction does indeed match the neural network's forward propagation, we consider an arbitrary latent node $H$ with $N$ unobserved parents

$h_1, ..., h_N$, and $M$ observed parents $g_1, ..., g_M$. The edges between these parents and $H$ then have weights $\theta_i$, $1 \leq i \leq N$, for the unobserved nodes, and weights $w_j$, $1 \leq j \leq M$, for the observed nodes. The network as a whole has observed evidence $\vec{x}$. For the rest of this problem we use a Markov network view of the neural network. The relevant potential passed from the unobserved parent nodes of $H$, $\phi(h_i)$ (as per the directed version of the graph) forward to $H$ have the following form:

| $h_i$ | $\neg h_i$ |
|-------|------------|
| $e^{p_i}$ | 1 |

Since $g_j$ are observed, their values are found in $\vec{x}$. Finally, the potentials between each of these nodes and the central node $H$ are as follows:

|  | $H$ | $\neg H$ |
|--|-----|----------|
| $h_i$ | $e^{\theta_i}$ | 1 |
| $\neg h_i$ | 1 | 1 |

|  | $H$ | $\neg H$ |
|--|-----|----------|
| $g_j$ | $e^{w_j}$ | 1 |
| $\neg g_j$ | 1 | 1 |

Suppose, then, using the second step of our construction, we make $L$ copies of all the nodes that were parents of $H$ in the directed version of the tree, $h_1^1, ..., h_1^L, ..., h_N^1, ..., h_N^L$ and $g_1^1, ..., g_1^L, ..., g_M^1, ..., g_M^L$ with weights $\theta_1/L, ..., \theta_N/L$ and $w_1/L, ..., w_M/L$ respectively. The potentials between $H$ and these copied nodes is then:

|  | $H$ | $\neg H$ |
|--|-----|----------|
| $h_i^k$ | $e^{\theta_i/L}$ | 1 |
| $\neg h_i^k$ | 1 | 1 |

|  | $H$ | $\neg H$ |
|--|-----|----------|
| $g_j^k$ | $e^{w_j/L}$ | 1 |
| $\neg g_j^k$ | 1 | 1 |

where $1 \leq i \leq N$, $1 \leq j \leq M$, and $1 \leq k \leq L$. The relevant potentials for each of the copied nodes to be passed forward are the same as the nodes they were originally copied from. We then have that,

$$\phi(H, h_1^1, ..., h_1^L, ..., h_N^1, ..., h_N^L, g_1^1, ..., g_1^L, ..., g_M^1, ..., g_M^L | \vec{x})$$

$$= \prod_{j=1}^{M} \prod_{k=1}^{L} e^{(w_j/L)H \times g_j^k} \times \prod_{i=1}^{N} \prod_{k=1}^{L} e^{(\theta_i/L)H \times h_i^k} \phi(h_i^k)$$

$$= e^{\sum_{j=1}^{M} w_j g_j \times H} \times \prod_{i=1}^{N} \prod_{k=1}^{L} e^{(\theta_i/L)H \times h_i^k} \phi(h_i^k).$$

Summing out an arbitrary, copied latent node, $h_\alpha^\beta$:

$$\sum_{h_\alpha^\beta, \neg h_\alpha^\beta} \phi(H, h_1^1, ..., h_1^L, ..., h_N^1, ..., h_N^L | \vec{x})$$

$$= e^{\sum_{j=1}^M w_j g_j \times H} \times \sum_{h_\alpha^\beta, \neg h_\alpha^\beta} \prod_{i=1}^N \prod_{k=1}^L e^{(\theta_i/L)H \times h_i^k} \phi(h_i^k)$$

$$= e^{\sum_{j=1}^M w_j g_j \times H} \times$$

$$\left[ e^{p_\alpha} e^{(\theta_\alpha/L)H} \times \prod_{\substack{i=1,..,N \\ (i,k)\neq(\alpha,\beta)}} \prod_{k=1,...L} e^{(\theta_i/L)H \times h_i^k} \phi(h_i^k) \right.$$

$$\left. + \prod_{\substack{i=1,..,N \\ (i,k)\neq(\alpha,\beta)}} \prod_{k=1,...L} e^{(\theta_i/L)H \times h_i^k} \phi(h_i^k) \right]$$

$$= e^{\sum_{j=1}^M w_j g_j \times H} \times (e^{p_\alpha} e^{(\theta_\alpha/L)H} + 1)$$

$$\times \left[ \prod_{\substack{i=1,..,N \\ (i,k)\neq(\alpha,\beta)}} \prod_{k=1,...L} e^{(\theta_i/L)H \times h_i^k} \phi(h_i^k) \right].$$

Summing out all $L$ copies of $h_\alpha$:

$$e^{\sum_{j=1}^M w_j g_j \times H} \times (e^{p_\alpha} e^{(\theta_\alpha/L)H} + 1)^L \times \left[ \prod_{\substack{i=1,..,N \\ i \neq \alpha}} \prod_{k=1,...L} e^{(\theta_i/L)H \times h_i^k} \phi(h_i^k) \right].$$

Then summing out the $L$ copies of each latent parent:

$$e^{\sum_{j=1}^M w_j g_j \times H} \times \prod_i^N (e^{p_i} e^{(\theta_i/L)H} + 1)^L ,$$

Normalizing this message locally, $\phi(H = 1 | \vec{x})$ becomes $\phi(H = 1 | \vec{x}) / \phi(H = 0 | \vec{x})$ and $\phi(H = 0 | \vec{x})$ becomes 1. This then gives us:

$$\phi(H = 1 | \vec{x})$$

$$= \left[ e^{\sum_{j=1}^M w_j g_j \times 1} \times \prod_i^N (e^{p_i} e^{(\theta_i/L) \times 1} + 1)^L \right] \Big/ \left[ e^{\sum_{j=1}^M w_j g_j \times 0} \times \prod_i^N (e^{p_i} e^{(\theta_i/L) \times 0} + 1)^L \right]$$

$$= \frac{e^{\sum_{j=1}^M w_j g_j} \times \prod_i^N (e^{p_i} e^{(\theta_i/L)} + 1)^L}{\prod_i^N (e^{p_i} + 1)^L}.$$

We then consider:

$$\lim_{L\to\infty} \frac{e^{\sum_{j=1}^{M} w_j g_j} \times \prod_i^N (e^{p_i} e^{(\theta_i/L)} + 1)^L}{\prod_i^N (e^{p_i} + 1)^L} =$$

$$\lim_{L\to\infty} \exp\left( \sum_{j=1}^{M} w_j g_j - \sum_{i=1}^{N} L \times \log(e^{p_i} + 1) \right.$$

$$\left. + \sum_{i=1}^{N} L \times \log(e^{p_i} e^{(\theta_i/L)} + 1) \right),$$

and the logarithm of this limit is,

$$\lim_{L\to\infty} \left[ \sum_{j=1}^{M} w_j g_j - \sum_{i=1}^{N} L \times \log(e^{p_i} + 1) \right.$$

$$\left. + \sum_{i=1}^{N} L \times \log(e^{p_i} e^{(\theta_i/L)} + 1) \right]$$

$$= \sum_{j=1}^{M} w_j g_j + \lim_{L\to\infty} \frac{\sum_{i=1}^{N} \left( \log(e^{p_i} e^{(\theta_i/L)} + 1) - \log(e^{p_i} + 1) \right)}{1/L}.$$

The limit in the previous expression clearly has the indeterminate form of $\frac{0}{0}$. Let $G = \sum_{j=1}^{M} w_j g_j$ and consider the following change of variables, $S = 1/L$, and subsequent use of l'Hôspital's rule.

$$G + \lim_{S\to 0^+} \frac{\sum_{i=1}^{N} \left( \log(e^{p_i} e^{(\theta_i S)} + 1) - \log(e^{p_i} + 1) \right)}{S}$$

$$= G + \lim_{S\to 0^+} \frac{\frac{\partial}{\partial S} \sum_{i=1}^{N} \left( \log(e^{p_i} e^{(\theta_i S)} + 1) - \log(e^{p_i} + 1) \right)}{\frac{\partial}{\partial S} S}$$

$$= G + \lim_{S\to 0^+} \frac{\sum_{i=1}^{N} \frac{1}{e^{p_i} e^{(\theta_i S)} + 1} \times e^{p_i} e^{\theta_i S} \times \theta_i}{1}$$

$$= G + \lim_{S\to 0^+} \sum_{i=1}^{N} \frac{e^{p_i} e^{\theta_i S} \times \theta_i}{e^{p_i} e^{\theta_i S} + 1}$$

$$= G + \sum_{i=1}^{N} \frac{e^{p_i}}{e^{p_i} + 1} \times \theta_i$$

$$= \sum_{j=1}^{M} w_j g_j + \sum_{i=1}^{N} \sigma(p_i) \theta_i.$$

Therefore,

$$\lim_{L\to\infty} \frac{e^{\sum_{j=1}^{M} w_j g_j} \times \prod_i^N (e^{p_i} e^{(\theta_i/L)} + 1)^L}{\prod_i^N (e^{p_i} + 1)^L}$$

$$= \exp\left( \sum_{j=1}^{M} w_j g_j + \sum_i^N \sigma(p_i) \theta_i \right),$$

The collected potential at node $H$ from summing out its ancestors (from the directed view) then has the form:

| $h_i$ | $\neg h_i$ |
|---|---|
| $\exp(\sum_{j=1}^{M} w_j g_j + \sum_{i}^{N} \sigma(p_i)\theta_i)$ | 1 |

This is exactly the form of messages that we assumed were originally passed to node $H$. Suppose then that $z$ is a hidden node whose parents in the original deep neural network's DAG are all observed. By our PGM construction, we have that node $z$ collects potential $e^{\sum_{x \in \vec{x}} w_{zx} x}$ for $z$ true and 1 for $z$ false from our initial forward step. Here $w_{zx}$ is the weight between nodes $z$ and $x$. Consider, then, the nodes whose parents in the DNN's DAG are either one of these first layer hidden nodes, or an observed node. By our PGM construction, we have shown that so long as the nodes in the previous layer are either observed or have this exponential message product, as is the case here, the message product of the nodes that immediately follow will have the same form.

Note that in order to calculate probability, $P(H|\vec{x})$, we must also consider the influence that the child of node $H$, call this $C$, has on node $H$ itself (this would be information passed through and collected at later nodes in the tree network that are then passed back through this node $C$ to node $H$) or may not exist at all in the case of output nodes. Suppose the weight between this child $C$ and node $H$ is $\gamma$. Suppose also that the message coming from node $C$ to $H$ has the following form, where $c$ is a non-negative real value that can be arbitrarily large.

| $C$ | $\neg C$ |
|---|---|
| $c$ | 1 |

Finally, note that with the $L$ copies made in this graph, the potential between node $H$ and $C$ has the form:

| | $H$ | $\neg H$ |
|---|---|---|
| $C$ | $e^{\gamma/L}$ | 1 |
| $\neg C$ | 1 | 1 |

Using the potential collected at $H$ from the forward pass and this addition information from $C$, we can then calculate the probability of node $H$ given the input evidence $\vec{x}$:

$$P(H|\vec{x}) = \frac{1}{Z} \times e^{(\sum_{j=1}^{M} w_j g_j + \sum_{i}^{N} \sigma(p_i)\theta_i) \times H} \times \sum_{C} e^{\gamma/L \times C \times H} \phi(C)$$

$$= \frac{1}{Z} \times e^{(\sum_{j=1}^{M} w_j g_j + \sum_{i}^{N} \sigma(p_i)\theta_i) \times H} \times (ce^{\gamma/L \times H} + 1)$$

From this we have that:

$$P(H = 1|\vec{x})$$
$$= \frac{e^{(\sum_{j=1}^{M} w_j g_j + \sum_{i}^{N} \sigma(p_i)\theta_i) \times 1} \times (ce^{\gamma/L \times C \times 1} + 1)}{e^{(\sum_{j=1}^{M} w_j g_j + \sum_{i}^{N} \sigma(p_i)\theta_i) \times 1} \times (ce^{\gamma/L \times C \times 1} + 1) + e^0 \times (ce^0 + 1)}$$
$$= \frac{e^{\sum_{j=1}^{M} w_j g_j + \sum_{i}^{N} \sigma(p_i)\theta_i} \times (ce^{\gamma/L \times C \times 1} + 1)}{e^{\sum_{j=1}^{M} w_j g_j + \sum_{i}^{N} \sigma(p_i)\theta_i} \times (ce^{\gamma/L \times C \times 1} + 1) + (c + 1)}$$
$$= \frac{e^{\sum_{j=1}^{M} w_j g_j + \sum_{i}^{N} \sigma(p_i)\theta_i}}{e^{\sum_{j=1}^{M} w_j g_j + \sum_{i}^{N} \sigma(p_i)\theta_i} + (c + 1)/(ce^{\gamma/L \times C \times 1} + 1)}.$$

Note that $\lim_{L \to \infty}(c + 1)/(ce^{\gamma/L \times C \times 1} + 1) = 1$, i.e. as $L$ grows increasingly large the information passed through this child node becomes negligible. We therefore have that $P(H = 1|\vec{x}) = \sigma(\sum_{j=1}^{M} w_j g_j + \sum_{i}^{N} \sigma(p_i)\theta_i)$, which is exactly the sigmoid activation value found in the forward pass of the neural network. Since every node in this network have the same form of incoming messages from their parents and child, we have that the conditional probability in this PGM construction and the activation values of the DNN match for any node in any layer of the DNN/PGM.

## D A Proof of the Gradient

From *An Introduction to Conditional Random Fields* Sutton and McCallum [2012] we have the gradient of weight $w_{pk}$ for this form of CRF can be written as:

$$\frac{\partial l}{\partial w_{pk}} = \sum_{\Psi_c \in C_p} \sum_{h'_c} P(h'_c|y, x) f_k(y_c, x_c, h'_c) - \sum_{\Psi_c \in C_p} \sum_{h'_c, y'_c} P(h'_c, y'_c|x) f_k(y'_c, x_c, h'_c).$$

Since the features on the cliques of the proposed infinite width-PGM structure are non-zero ($1/L$, which is a minor change from instead dividing each weight by $L$) only when the two adjacent nodes are both 1 (true), this expression simplifies. The update on a specific edge in the PGM then takes the following form and the complete weight update would be the summation over the updates on each edge the weight of interest appears.

$$(1/L)[P(h_n = 1, h_{n+1} = 1|x, y) - P(h_n = 1, h_{n+1} = 1|x)],$$

where $h_n$ and $h_{n+1}$ are the nodes that the edge connects.

Consider a single branch in the infinite width PGM structure. Let $h_n$ be the node closer to the output node and let it be exactly $n$ nodes separated from said output (node $h_{n+1}$ is then naturally the next node in this branch). Due to the infinite-width structure of the proposed PGM there are $L^{n+1}$ copies of this exact path ending in the weight of interest. These copied paths are entirely equivalent to one another.

Suppose now we sum out the influence output $y$ has on the weight update of these paths. Note that this could be written equivalently as summing out node $h_0$.

$$
\begin{aligned}
&P(h_n = 1, h_{n+1} = 1|x, y) - P(h_n = 1, h_{n+1} = 1|x) \\
&= P(h_n = 1, h_{n+1} = 1|x, y) - \sum_y P(h_n = 1, h_{n+1} = 1, y|x) \\
&= P(h_n = 1, h_{n+1} = 1|x, y) - \sum_y P(h_n = 1, h_{n+1} = 1|x, y) P(y|x) \\
&= y P(h_n = 1, h_{n+1} = 1|x, y = 1) + (1 - y) P(h_n = 1, h_{n+1} = 1|x, y = 0) \\
&\quad - P(h_n = 1, h_{n+1} = 1|x, y = 1) P(y = 1|x) \\
&\quad - P(h_n = 1, h_{n+1} = 1|x, y = 0) P(y = 0|x) \\
&= (y - \hat{y}) P(h_n = 1, h_{n+1} = 1|x, y = 1) \\
&\quad + (1 - y - (1 - \hat{y})) P(h_n = 1, h_{n+1} = 1|x, y = 0) \\
&= (y - \hat{y})[P(h_n = 1, h_{n+1} = 1|x, y = 1) - P(h_n = 1, h_{n+1} = 1|x, y = 0)]
\end{aligned}
$$

Note that if we are considering the weight update on the connections between the output and its neighbors, this update can be used immediately as $y$ would be $h_0$. The update across all $L$ copies would then have the form:

$$
\begin{aligned}
&\sum_{k=1}^{L} (1/L)(y - \hat{y})[P(h_0 = 1, h_1 = 1|x, h_0 = 1) - P(h_0 = 1, h_1 = 1|x, h_0 = 0)] \\
&= \sum_{k=1}^{L} (1/L)(y - \hat{y}) P(h_1 = 1|x, h_0 = 1) \\
&= (y - \hat{y}) P(h_1 = 1|x, h_0 = 1)
\end{aligned}
$$

We can easily calculate $P(h_1 = 1|x, h_0 = 1)$ used in this update as follows. Let node $h_i$ have conditional probability $\sigma(S_{h_i})$, which corresponds to a forward product of messages of the form:

| $h_i$ | $\neg h_i$ |
|-------|-----------|
| $e^{S_{h_i}}$ | 1 |

Let the weight between nodes $h_i$ and $h_{i-1}$ in the path of interest be $w_{i-1}$. The product of messages at $h_i$ then including the fact that we know $h_{i-1} = 1$, then has the form:

| $h_i$ | $\neg h_i$ |
|---|---|
| $e^{S_{h_i} + w_{i-1}/L}$ | 1 |

From this, we can then calculate:

$$P(h_i = 1 | h_{i-1} = 1) = \sigma(S_{h_i} + w_{i-1}/L). \tag{D.1}$$

For the update of the weights surrounding the output, the overall update must then have the form $(1/L)(y - \hat{y})\sigma(S_{h_1} + w_0/L)$. Note that there are $L$ such copies of this update and $L$ is infinitely large. The overall update of $w_0$ is then finally, $\lim_{L\to\infty} \sum_{k=1}^{L} (1/L)(y-\hat{y})\sigma(S_{h_1} + w_0/L) = (y-\hat{y})\sigma(S_{h_1})$. This exactly matches the update of $w_0$ in the gradient descent step of the neural network.

We now consider the updates of weights with further depth away from the output. This update take the form $(1/L)(y - \hat{y})[P(h_n = 1, h_{n+1} = 1 | x, y = 1) - P(h_n = 1, h_{n+1} = 1 | x, y = 0)]$. For the remainder of this proof we focus on this difference of probabilities. Note however, the complete update does include this $(1/L)(y - \hat{y})$ term.

$$
\begin{aligned}
&P(h_n = 1, h_{n+1} = 1 | x, y = 1) - P(h_n = 1, h_{n+1} = 1 | x, y = 0) \\
&= \sum_{h_1} [P(h_n = 1, h_{n+1} = 1, h_1 | x, y = 1) - P(h_n = 1, h_{n+1} = 1, h_1 | x, y = 0)] \\
&= P(h_n = 1, h_{n+1} = 1, h_1 = 1 | x, y = 1) \\
&\quad - P(h_n = 1, h_{n+1} = 1, h_1 = 1 | x, y = 0) \\
&\quad + P(h_n = 1, h_{n+1} = 1, h_1 = 0 | x, y = 1) \\
&\quad - P(h_n = 1, h_{n+1} = 1, h_1 = 0 | x, y = 0) \\
&= P(h_n = 1, h_{n+1} = 1 | x, y = 1, h_1 = 1) P(h_1 = 1 | x, y = 1) \\
&\quad - P(h_n = 1, h_{n+1} = 1 | x, y = 0, h_1 = 1) P(h_1 = 1 | x, y = 0) \\
&\quad + P(h_n = 1, h_{n+1} = 1 | x, y = 1, h_1 = 0) P(h_1 = 0 | x, y = 1) \\
&\quad - P(h_n = 1, h_{n+1} = 1 | x, y = 0, h_1 = 0) P(h_1 = 0 | x, y = 0) \qquad \text{(Since } h_n, h_{n+1} \perp y | h_1) \\
&= P(h_n = 1, h_{n+1} = 1 | x, h_1 = 1) P(h_1 = 1 | x, y = 1) \\
&\quad - P(h_n = 1, h_{n+1} = 1 | x, h_1 = 1) P(h_1 = 1 | x, y = 0) \\
&\quad + P(h_n = 1, h_{n+1} = 1 | x, h_1 = 0) P(h_1 = 0 | x, y = 1) \\
&\quad - P(h_n = 1, h_{n+1} = 1 | x, h_1 = 0) P(h_1 = 0 | x, y = 0) \\
&= P(h_n = 1, h_{n+1} = 1 | x, h_1 = 1)(P(h_1 = 1 | x, y = 1) - P(h_1 = 1 | x, y = 0)) \\
&\quad + P(h_n = 1, h_{n+1} = 1 | x, h_1 = 0)(P(h_1 = 0 | x, y = 1) - P(h_1 = 0 | x, y = 0)) \\
&= [P(h_n = 1, h_{n+1} = 1 | x, h_1 = 1) \\
&\quad \times (P(h_1 = 1 | x, y = 1) - P(h_1 = 1 | x, y = 0))] \\
&\quad + [P(h_n = 1, h_{n+1} = 1 | x, h_1 = 0) \\
&\quad \times (1 - P(h_1 = 1 | x, y = 1) - (1 - P(h_1 = 1 | x, y = 0)))] \\
&= (P(h_1 = 1 | x, y = 1) - P(h_1 = 1 | x, y = 0)) \times \\
&\quad [P(h_n = 1, h_{n+1} = 1 | x, h_1 = 1) - P(h_n = 1, h_{n+1} = 1 | x, h_1 = 0)]
\end{aligned}
$$

Note that $P(h_n = 1, h_{n+1} = 1 | x, h_1 = 1) - P(h_n = 1, h_{n+1} = 1 | x, h_1 = 0)$ has the exact same form as the difference we started with, but $y$ (or written equivalently $h_0$) has been effectively replaced with $h_1$. Due to the tree structure of the PGM, the conditional independence relationships are also entirely equivalent. If $h_2$ then also existed in this branch and preceded $h_n$, we could then apply the exact same operations to 'sum out' $h_1$ in this expression and replace it with $h_2$. This can be repeated until $h_n$ itself is reached and we condition on the node one position closer to the output $h_{n+1}$. Note that at for step we must multiply the entire expression by $(P(h_i = 1 | x, h_{i-1} = 1) - P(h_i = 1 | x, h_{i-1} = 0))$ terms.

Suppose, for the sake of induction,

$$P(h_n = 1, h_{n+1} = 1 | x, y = 1) - P(h_n = 1, h_{n+1} = 1 | x, y = 0)$$
$$= [\prod_{i=1}^{k}(P(h_i = 1 | x, h_{i-1} = 1) - P(h_i = 1 | x, h_{i-1} = 0))]$$
$$\times [P(h_n = 1, h_{n+1} = 1 | x, h_k = 1) - P(h_n = 1, h_{n+1} = 1 | x, h_k = 0)],$$

where $n > k$. We have already shown that this holds for the base case of $k = 1$. We then consider the case of $(k + 1)$:

$$P(h_n = 1, h_{n+1} = 1 | x, y = 1) - P(h_n = 1, h_{n+1} = 1 | x, y = 0) \quad \text{(By the inductive hypothesis)}$$
$$= [\prod_{i=1}^{k}(P(h_i = 1 | x, h_{i-1} = 1) - P(h_i = 1 | x, h_{i-1} = 0))]$$
$$\times [P(h_n = 1, h_{n+1} = 1 | x, h_k = 1) - P(h_n = 1, h_{n+1} = 1 | x, h_k = 0)]$$
$$= [\prod_{i=1}^{k}(P(h_i = 1 | x, h_{i-1} = 1) - P(h_i = 1 | x, h_{i-1} = 0))]$$
$$\times \sum_{h_{k+1}}[P(h_n = 1, h_{n+1} = 1, h_{k+1} | x, h_k = 1) - P(h_n = 1, h_{n+1} = 1, h_{k+1} | x, h_k = 0)]$$
$$= [\prod_{i=1}^{k}(P(h_i = 1 | x, h_{i-1} = 1) - P(h_i = 1 | x, h_{i-1} = 0))]$$
$$\times \sum_{h_{k+1}}[P(h_n = 1, h_{n+1} = 1 | x, h_k = 1, h_{k+1})P(h_{k+1} | x, h_k = 1)$$
$$- P(h_n = 1, h_{n+1} = 1 | x, h_k = 0, h_{k+1})P(h_{k+1} | x, h_k = 0)] \quad \text{(Since } h_n, h_{n+1} \perp h_k | h_{k+1})$$
$$= [\prod_{i=1}^{k}(P(h_i = 1 | x, h_{i-1} = 1) - P(h_i = 1 | x, h_{i-1} = 0))]$$
$$\times \sum_{h_{k+1}}[P(h_n = 1, h_{n+1} = 1 | x, h_{k+1})P(h_{k+1} | x, h_k = 1)$$
$$- P(h_n = 1, h_{n+1} = 1 | x, h_{k+1})P(h_{k+1} | x, h_k = 0)]$$
$$= [\prod_{i=1}^{k}(P(h_i = 1 | x, h_{i-1} = 1) - P(h_i = 1 | x, h_{i-1} = 0))]$$
$$\times \sum_{h_{k+1}}[P(h_n = 1, h_{n+1} = 1 | x, h_{k+1})(P(h_{k+1} | x, h_k = 1) - P(h_{k+1} | x, h_k = 0))]$$
$$= [\prod_{i=1}^{k}(P(h_i = 1 | x, h_{i-1} = 1) - P(h_i = 1 | x, h_{i-1} = 0))]$$
$$\times [P(h_n = 1, h_{n+1} = 1 | x, h_{k+1} = 1)(P(h_{k+1} = 1 | x, h_k = 1) - P(h_{k+1} = 1 | x, h_k = 0))$$
$$+ P(h_n = 1, h_{n+1} = 1 | x, h_{k+1} = 0)(P(h_{k+1} = 0 | x, h_k = 1) - P(h_{k+1} = 0 | x, h_k = 0))]$$
$$= [\prod_{i=1}^{k}(P(h_i = 1 | x, h_{i-1} = 1) - P(h_i = 1 | x, h_{i-1} = 0))]$$
$$\times [P(h_n = 1, h_{n+1} = 1 | x, h_{k+1} = 1)(P(h_{k+1} = 1 | x, h_k = 1) - P(h_{k+1} = 1 | x, h_k = 0))$$
$$+ P(h_n = 1, h_{n+1} = 1 | x, h_{k+1} = 0)$$
$$\times (1 - P(h_{k+1} = 1 | x, h_k = 1) - 1 + P(h_{k+1} = 1 | x, h_k = 0))]$$

$$= [\prod_{i=1}^{k}(P(h_i = 1|x, h_{i-1} = 1) - P(h_i = 1|x, h_{i-1} = 0))]$$
$$\times [P(h_n = 1, h_{n+1} = 1|x, h_{k+1} = 1)(P(h_{k+1} = 1|x, h_k = 1) - P(h_{k+1} = 1|x, h_k = 0))$$
$$- P(h_n = 1, h_{n+1} = 1|x, h_{k+1} = 0)(P(h_{k+1} = 1|x, h_k = 1) - P(h_{k+1} = 1|x, h_k = 0))]$$
$$= [\prod_{i=1}^{k+1}(P(h_i = 1|x, h_{i-1} = 1) - P(h_i = 1|x, h_{i-1} = 0))]$$
$$\times [P(h_n = 1, h_{n+1} = 1|x, h_{k+1} = 1) - P(h_n = 1, h_{n+1} = 1|x, h_{k+1} = 0)].$$

We therefore have that $P(h_n = 1, h_{n+1} = 1|x, y = 1) - P(h_n = 1, h_{n+1} = 1|x, y = 0)$ can be written as a product of the differences $P(h_i = 1|x, h_{i-1} = 1) - P(h_i = 1|x, h_{i-1} = 0)$ multiplied by the difference $P(h_n = 1, h_{n+1} = 1|x, h_k = 1) - P(h_n = 1, h_{n+1} = 1|x, h_k = 0)$, for any $k < n$.

Note that from D.1 we have that $P(h_i = 1|x, h_{i-1} = 1) = \sigma(S_{h_i} + w_{i-1}/L)$ where $w_{i-1}$ is the weight between the two nodes and $S_{h_i}$ is the internal summation at $h_i$ from the forward pass of the neural network prior to applying the sigmoid activation function $\sigma(\cdot)$. It similarly follows that $P(h_i = 1|x, h_{i-1} = 0) = \sigma(S_{h_i})$. We then divide and multiply this difference by $w_{i-1}/L$.

$$P(h_i = 1|x, h_{i-1} = 1) - P(h_i = 1|x, h_{i-1} = 0))$$
$$= (w_{i-1}/L)\frac{P(h_i = 1|x, h_{i-1} = 1) - P(h_i = 1|x, h_{i-1} = 0))}{w_{i-1}/L}$$
$$= (w_{i-1}/L)\frac{\sigma(S_{h_i} + w_{i-1}/L) - \sigma(S_{h_i})}{w_{i-1}/L}$$

Note that the right hand term of the above expression has the same form as the definition of a derivative when placed within the limit as $L$ approaches infinity, i.e.:

$$\lim_{L\to\infty}\frac{\sigma(S_{h_i} + w_{i-1}/L) - \sigma(S_{h_i})}{w_{i-1}/L} = \frac{\partial}{\partial S_{h_i}}\sigma(S_{h_i}).$$

At each step away from the output of the network along this branch, we are therefore multiplying by derivative of the local sigmoid per step (this exactly matches the behaviour of the backward pass of the neural network) multiplied by the relevant weight. There is naturally the concern of the remaining $\frac{1}{L}$ term both per step and in the $(y - \hat{y})/L$ term. Recall that there are $L^{n+1}$ copies of the weight of interest in branches with the exact same structure. We sum over these equivalent weight updates and so have $\sum_{i=1}^{L^{n+1}}\frac{1}{L^{n+1}} = 1$. The summation of all these equivalent weight updates effectively cancels this repeated division (note that this is including the $1/L$ term shown in the final step of this proof).

We finally consider the difference involving nodes $n$ and $n + 1$ conditioned on node $n - 1$.

$$P(h_n = 1, h_{n+1} = 1|x, h_{n-1} = 1) - P(h_n = 1, h_{n+1} = 1|x, h_{n-1} = 0)$$
$$= P(h_{n+1} = 1|x, h_{n-1} = 1, h_n = 1)P(h_n = 1|x, h_{n-1} = 1)$$
$$- P(h_{n+1} = 1|x, h_{n-1} = 0, h_n = 1)P(h_n = 1|x, h_{n-1} = 0) \quad (\text{Since } h_{n+1} \perp h_{n-1}|h_n)$$
$$= P(h_{n+1} = 1|x, h_n = 1)P(h_n = 1|x, h_{n-1} = 1)$$
$$- P(h_{n+1} = 1|x, h_n = 1)P(h_n = 1|x, h_{n-1} = 0)$$
$$= \sigma(S_{n+1} + w_n/L) \times (P(h_n = 1|x, h_{n-1} = 1) - P(h_n = 1|x, h_{n-1} = 0))$$
$$= \sigma(S_{n+1} + w_n/L) \times (w_{n-1}/L)(\frac{\sigma(S_{h_n} + w_{n-1}/L) - \sigma(S_{h_n})}{w_{n-1}/L})$$

The right hand term becomes the derivative of $\sigma(S_{h_n})$ as in earlier steps and the left $\sigma(\cdot)$ term becomes $\sigma(S_{n+1})$ i.e. the forward pass value at node $n + 1$, the last node relevant to this weight calculation on this branch.

The entire weight update then has the form:

$$(y - \hat{y}) \times [\prod_{i=1}^{n} w_{i-1} \frac{\partial}{\partial S_{h_i}} \sigma(S_{h_i})] \times \sigma(S_{n+1})$$

This is exactly the gradient update in the neural network when considering a specific path/weight combination in the network. The infinite width PGM naturally has branches for each possible path in the neural network and as such the complete weight updates of both views will align.

## E   HMC Sampling Trajectories

Suppose the current chain state is $\boldsymbol{h}^{(n)} = \boldsymbol{\rho}_n(0)$. We then draw a momentum $\boldsymbol{\mu}_n(0) \sim \mathcal{N}(0, M)$. The HMC trajectories imply that after $\Delta t$, we have:

$$\boldsymbol{\mu}_n(t + \frac{\Delta t}{2}) = \boldsymbol{\mu}_n(t) - \frac{\Delta t}{2} \nabla U(\boldsymbol{\rho}) \Big|_{\boldsymbol{\rho} = \boldsymbol{\rho}_n(t)}$$

$$\boldsymbol{\rho}_n(t + \Delta t) = \boldsymbol{\rho}_n(t) + \Delta t M^{-1} \boldsymbol{\mu}_n(t + \frac{\Delta t}{2}) \tag{E.1}$$

$$\boldsymbol{\mu}_n(t + \Delta t) = \boldsymbol{\mu}_n(t + \frac{\Delta t}{2}) - \frac{\Delta t}{2} \nabla U(\boldsymbol{\rho}) \Big|_{\boldsymbol{\rho} = \boldsymbol{\rho}_n(t + \Delta t)} .$$

We may then apply these equations to $\boldsymbol{\rho}_n(0)$ and $\boldsymbol{\mu}_n(0)$ $L$ times to get $\boldsymbol{\rho}_n(L\Delta t)$ and $\boldsymbol{\mu}_n(L\Delta t)$. Thus, the transition from $\boldsymbol{h}_{(n)} = \boldsymbol{\rho}_n$ to the next state $\boldsymbol{h}^{(n+1)}$ is given by:

$$\boldsymbol{h}^{(n+1)} \Big| (\boldsymbol{h}^{(n)} = \boldsymbol{\rho}_n(0)) = \begin{cases} \boldsymbol{\rho}_n(L\Delta t) & \text{with probability } \alpha(\boldsymbol{\rho}_n(0), \boldsymbol{\rho}_n(L\Delta t)) \\ \boldsymbol{\rho}_n(0) & \text{otherwise} \end{cases} \tag{E.2}$$

where

$$\alpha(\boldsymbol{\rho}_n(0), \boldsymbol{\rho}_n(L\Delta t)) = \min \left( 1, \exp \left( H(\boldsymbol{\rho}_n(0), \boldsymbol{\mu}_n(0)) - H(\boldsymbol{\rho}_n(L\Delta t), \boldsymbol{\mu}_n(L\Delta t)) \right) \right). \tag{E.3}$$

## F   CD-k Algorithm

---

**Algorithm 3** CD-$k$ Learning for the Deep Belief Network

---

**Input:** Initialized $\boldsymbol{h}^{(0)}$, $\boldsymbol{W}^{(0)} = \{\boldsymbol{W}_i^{(0)} | i = 1, 2, ..., K\}$, $\boldsymbol{b}^{(0)} = \{\boldsymbol{b}_i^{(0)} | i = 1, 2, ..., K\}$
**Output:** $\boldsymbol{W}^{(n)}, \boldsymbol{b}^{(n)}$ when loss converges.

1: **procedure** BURN-IN($N$)
2:     **for** $i \leftarrow 0$ to $N - 1$ **do**
3:         $\boldsymbol{h}^{(i+1)} \leftarrow \text{Sampling}(\boldsymbol{h}^{(i)}, \boldsymbol{W}^{(0)}, \boldsymbol{b}^{(0)})$
4:     **end for**
5: **end procedure**
6: **procedure** TRAINING($M$)
7:     **for** $i \leftarrow 0$ to $M - 1$ **do**
8:         $\boldsymbol{W}^{(i+1)}, \boldsymbol{b}^{(i+1)} \leftarrow \text{Weight-updating}(\boldsymbol{h}^{(N+ik)}, \boldsymbol{W}^{(i)}, \boldsymbol{b}^{(i)})$
9:         **for** $j \leftarrow 0$ to $k - 1$ **do**
10:           $\boldsymbol{h}^{(N+ik+j+1)} \leftarrow \text{Sampling}(\boldsymbol{h}^{(N+ik+j)}, \boldsymbol{W}^{(i+1)}, \boldsymbol{b}^{(i+1)})$
11:         **end for**
12:     **end for**
13: **end procedure**

---

In our experiments, $k = 1$, i.e. we do one sampling step before weight updating. We mainly use HMC sampling method for the sampling step, where the detailed trajectories are defined in Appendix E. The weight-updating step is defined in equations (2), (6) and (7).

We also run Gibbs method for comparison, where results are shown in Table 1. Both Gibbs and HMC are sampling methods to generate new states for all the hidden nodes and they both apply CD-1 or CD-k to learn the weight. While HMC samples real values in the range of $[0, 1]$ under the normal distribution, Gibbs samples values in $\{0, 1\}$ under Bernoulli distribution. For node $h_{ij}$, i.e., the $j$th node in $\boldsymbol{h}_i$ layer, the Bernoulli distribution used to sample its new state is determined by its Markov blanket, which includes all the nodes in $\boldsymbol{h}_{i-1}, \boldsymbol{h}_i,$ and $\boldsymbol{h}_{i+1}$ in this case. It could be calculated by normalizing the joint probability distribution of the blanket when $h_{ij} = 1$ versus $h_{ij} = 0$, which is written as following:

$$p(h_{ij} = 1) = \frac{p(h_{ij} = 1|\boldsymbol{h}_{i-1}) \cdot p(\boldsymbol{h}_{i+1}|\boldsymbol{h}_i\backslash\{h_{ij}\}, h_{ij} = 1)}{p(h_{ij} = 0|\boldsymbol{h}_{i-1}) \cdot p(\boldsymbol{h}_{i+1}|\boldsymbol{h}_i\backslash\{h_{ij}\}, h_{ij} = 0) + p(h_{ij} = 1|\boldsymbol{h}_{i-1}) \cdot p(\boldsymbol{h}_{i+1}|\boldsymbol{h}_i\backslash\{h_{ij}\}, h_{ij} = 1)}$$

We then sample a new state for $h_{ij}$ from $\text{Bern}(p(h_{ij} = 1))$. In Gibbs sampling, the node is sampled one by one since a newly sampled node will affect the sampling of subsequent nodes in its Markov blanket. After sampling new values for all the hidden nodes, the weight is similarly updated using equations (2), (6) and (7). The Gibbs sampling does not depend on $L$.

## G  Experimental Setup

For all the experiments, the train-test split ratio is 80:20. For the training and finetuning, Adam optimizer is used with learning rate being $1 \times 10^{-4}$. To get the predicted probabilities for fine-tuned network, 1000 output probabilities are sampled and averaged. In synthetic experiments, both BNs and MNs have the structure with the input dimension being 4, two latent layers with 4 nodes in each one, and one binary output.

## H  Running time

Running time of one BN experiment in the main text are shown in Table 3.

Table 3: Average running time of different methods on synthetic datasets. For DNN, it shows the time of the training for 100 and 1000 epochs. For Gibbs and HMC, it shows the time of the finetuning for 20 epochs.

| Data (Weight) | # Train Epochs | Average Running Time (s) | | | | |
|---|---|---|---|---|---|---|
| | | DNN | Gibbs | HMC-10 | HMC-100 | HMC-1000 |
| BN (0.3) | 100 | 7.07 | 2769.81 | 666.33 | 650.10 | 758.37 |
| | 1000 | 60.31 | 2741.30 | 634.20 | 626.79 | 728.57 |
| BN (1) | 100 | 7.22 | 2811.77 | 640.68 | 661.12 | 757.21 |
| | 1000 | 65.25 | 2784.53 | 617.21 | 643.96 | 730.81 |
| BN (3) | 100 | 5.09 | 2616.04 | 615.96 | 615.05 | 636.93 |
| | 1000 | 44.83 | 2602.65 | 596.23 | 594.62 | 608.02 |
| BN (10) | 100 | 6.87 | 2686.98 | 638.80 | 665.14 | 661.82 |
| | 1000 | 48.98 | 2666.23 | 617.72 | 646.08 | 646.06 |
| MN (0.3) | 100 | 7.26 | 2703.71 | 645.57 | 635.42 | 729.60 |
| | 1000 | 62.38 | 2669.61 | 625.38 | 618.07 | 701.31 |
| MN (1) | 100 | 4.94 | 2508.05 | 578.88 | 582.59 | 610.07 |
| | 1000 | 44.39 | 2474.70 | 559.11 | 560.54 | 584.33 |
| MN (3) | 100 | 10.84 | 2610.14 | 591.43 | 579.26 | 605.13 |
| | 1000 | 52.82 | 2578.81 | 570.23 | 558.98 | 585.22 |

