# OpenReview forum: "On Neural Networks as Infinite Tree-Structured Probabilistic Graphical Models"
_NeurIPS.cc/2024/Conference — NeurIPS 2024 poster_

### Official Review · Reviewer_ymC3 · 2024-07-05

**Soundness:** 3
**Presentation:** 2
**Contribution:** 2
**Rating:** 5
**Confidence:** 4

**Summary:**

The paper "On Neural Networks as Infinite Tree-Structured Probabilistic Graphical Models" introduces a framework that constructs infinite tree-structured probabilistic graphical models (PGMs) corresponding to deep neural networks (DNNs), demonstrating that DNNs perform precise approximations of PGM inference during forward propagation.
While this claim, along with others in the paper, sounds plausible, it also seems well-known within the field. The transition from Fig. 1a (Neural Network represented graphically) to the tree-like graphical model shown in Fig. 1b appears to simply construct what is called a computational tree in the graphical model literature. For reference, see several earlier papers discussing the topic:
•	Sekhar C. Tatikonda and Michael I. Jordan. "Loopy belief propagation and Gibbs measures." Proceedings of the Eighteenth Conference on Uncertainty in Artificial Intelligence, UAI’02, page 493–500, San Francisco, CA, USA, 2002. Morgan Kaufmann Publishers Inc.
•	Alexander T. Ihler, John W. Fisher III, and Alan S. Willsky. "Loopy belief propagation: Convergence and effects of message errors." Journal of Machine Learning Research, 6(31):905–936, 2005.
•	Dror Weitz. "Counting independent sets up to the tree threshold." Proceedings of the Thirty-Eighth Annual ACM Symposium on Theory of Computing, STOC ’06, page 140–149, New York, NY, USA, 2006. Association for Computing Machinery.
The last paper suggests an exact map of a finite loopy graphical model of the Markov Random field type into a large but finite loopless graphical model (tree). The infinite graphical model (computational tree) is linked to approximate inference (with belief propagation) in the original graphical model.
Computational tree-based approaches have previously been associated with both variational and MCMC techniques, though they were not developed into practical algorithms.
I find the Hamiltonian Monte Carlo algorithm developed in the paper, along with the reported experiments, quite interesting. However, I believe that more experiments and exploration are needed, as well as improved references to prior work in graphical models, before the paper is ready for publication.

**Strengths:**

see summary

**Weaknesses:**

see summary

**Questions:**

see summary

**Limitations:**

see summary

---

> ### Author Rebuttal · Authors · 2024-08-07
>
> We thank the reviewer for this very interesting comment about the computation tree and loopy BP.   While the computation tree is certainly a useful tool for precisely characterizing the approximation made by loopy belief propagation, we do not believe that the construction presented in this work is a trivial extension of that idea. Whereas belief propagation is exact in tree structured (non-loopy) graphs, the forward pass of the neural network when compared to conditional probability values of the Markov network with the potentials described in this work will not match even in the simple case of several hidden nodes connected in a line. Unlike loopy belief propagation then, which is well defined probabilistically in these simple cases, the neural network's forward operation must be captured by the infinite copies defined by the second step of our construction. This already differs rather notably from the use of the computation tree to characterize loopy belief propagation's approximation as that theoretical approach involves designing a tree whose values when computed are ordered correctly such that the influence of the loopy updates of belief propagation in non-tree structured graphs on nodes updated later are properly captured.
>
> Since the neural network's forward pass does not pass information backward, or in loops, the ordering of the nodes in our construction (ignoring the copies at this point) is wholly different from the computation tree that would be derived from using loopy belief propagation in a similarly structured starting graph. For example, in Figure 1 (page 3) of Loopy Belief Propagation and Gibbs Measures by Tatikonda and Jordan, it can be seen that when calculating the probability of node $a$ using the computation tree, an earlier belief about node $a$ is used earlier in the tree. For the neural network forward pass, however, information is never passed from a node back to itself. As such, the corresponding construction only ever creates copies of nodes in order to create a tree structure and account for how a local expectation is passed forward. It never creates a reordering of said nodes (as they would be ordered within the layers of the neural network). Belief propagation will naturally provide the same conditional probability as the forward pass of the neural network when it is applied within the infinite-width tree structured PGM, but that is simply a result of belief propagation being exact in tree structured graphs and said construction matching the operations of the neural network. When applied to the initial, loopy, graph defined by the neural network's structure, the relevant computation tree would not properly capture the neural network's operations.

---

> ### Comment · Reviewer_ymC3 · 2024-08-13
>
> In response to the authors' rebuttal, I want to clarify that I am not suggesting their work is a trivial extension of the computational tree framework. However, the computational tree serves as a significant reference point that should be discussed in the final version, should the paper be accepted.
>
> To further clarify my comments:
>
> 1) My observations regarding the relationship to the computational tree specifically pertain to the forward propagation aspect of the algorithm, where, as I understand it, the structure of the candidate tree and factors is fixed.
>
> 2) The approach introduced by Dror Weitz (as mentioned in my original review) constructs not an infinite but a specifically truncated portion of the computational tree. This construction provides exact, rather than approximate, inference for the known Graphical Model, particularly in the forward part of the algorithm discussed by the authors.

---

> ### Author Response · Authors · 2024-08-13
>
> Thank you for the clarification. We agree that the work that has been done using computation trees warrants mentioning in reference to the proposed construction and we will include a corresponding discussion in the final version of the paper.
>
> (1) The structure of the candidate tree and factors are fixed, which certainly aligns with how copies of nodes are added in the computation tree, even if the ordering of those nodes differs.
>
> (2) The truncated tree presented in Dror Weitz's work certainly appears most similar to the first step of our PGM construction due to its finite nature and perhaps an easier way to present that first step would be to describe it as consisting of several paths (walks) that follow the direction of the arrows in the initial neural network DAG that are copied in such a way that they do not overlap (and where copies of a node never appear as ancestors to the same node). That said, we would be cautious to very closely align the two as the first step of our construction follows this directed/layer-by-layer ordering while the graph presented in Weitz's paper is undirected.
>
> We thank the reviewer again for bringing our attention to this interesting connection.

---

### Official Review · Reviewer_BE6o · 2024-07-12

**Soundness:** 3
**Presentation:** 3
**Contribution:** 3
**Rating:** 6
**Confidence:** 3

**Summary:**

The paper proposes a novel connection between DNNs and PGMs. Specifically, the idea s to unroll the DNN computation graph in the form of a PGM. The main formal result that is shown is that DNN forward propagation corresponds to exact inference in the encoded PGM. More practically, since the PGM is infinite, it should not be expected to replace DNN-based operations, but rather to understand the working of a DNN from a PGM perspective. Further, a new HMC algorithm is developed using this view and the resulting approach can be used model calibration. Experiments are performed on synthetic data from a known distribution to measure the calibration error. Another set of experiments are performed with Covertype datasets

**Strengths:**

- The paper presents a nice connection between exact inference in PGMs with DNN propagation (for sigmoid units) which seems to be previously unknown
- There could be many foundational results that may be possible with the proposed connection (e.g. approximate inference in PGMs, etc.)
- The paper is well written and makes its contributions and limitations very clear

**Weaknesses:**

- The main weakness is perhaps in that the practical utility of going from the DNN function space to the PGM space is yet unknown. The experiments on calibration seem to show some promise but maybe stronger real-world studies could have demonstrated the need for the connection.

**Questions:**

If inference in a PGM corresponds to DNN propagation, then what is the likely source of miscalibration in the DNN models, I was wondering there is some way of knowing this using the proposed connection.

**Limitations:**

Limitations are identified and adequately discussed.

---

> ### Author Rebuttal · Authors · 2024-08-07
>
> Thank you for the great comment and question. The weakness is addressed in the general response. Regarding your question, we suspect that deep neural network miscalibration may arise from the non-standard infinite width structure of the proposed probabilistic graphical model (PGM). While inference in the infinite width PGM matches the forward operations of the neural network, it is itself, a remarkably unwieldy structure. This can be seen perhaps most easily by thinking of the flow of information across the neural network during its forward pass, how, if the value of a hidden neuron is observed, i.e. no longer hidden, that information does not affect earlier neurons (unlike in more traditional PGM structures where that information might be passed backward). The infinite copies of each node in the PGM construction of the neural network also effectively passes a local average value of each parent node forward, which is itself quite restrictive.
>
> Therefore, it seem possible that graphs more closely aligned to traditional Markov Random Fields (MRFs) or Bayesian networks might be less prone to the same types of miscalibration found in neural networks when the relationship between the inputs/outputs is less deterministic. The experiments with smaller choices of $L$ were done with the aim of potentially bridging the gap between the infinite width model and a less extreme tree structured PGM. Clearly when the input/output relationship is largely deterministic, the neural network can simply match its input/output predictions to high confidence values due to being a universal function approximator, but in settings with a more probabilistic nature this disconnect between the underlying infinite-width tree structure PGM that represents the DNN's operations and a, perhaps, more plausible, traditional PGMs could be a source of under/over-confidence.

---

> > ### Comment · Reviewer_BE6o · 2024-08-13
> >
> > Thanks for your response. I think the connection between PGMs and DNNs can be beneficial in several ways. I think this area is worth exploring and revisiting (including connections to some of the older works that other reviewers have mentioned). I am in general positive about this work.

---

### Official Review · Reviewer_7PML · 2024-07-13

**Soundness:** 3
**Presentation:** 2
**Contribution:** 4
**Rating:** 7
**Confidence:** 3

**Summary:**

This work bridges the gap between Deep Neural Networks (DNNs) and Probabilistic Graphical Models (PGMs). The authors do so by viewing DNNs as defining a joint distribution over the values of their nodes and showing that forward pass on a DNN is equivalent to exact probabilistic inference on a corresponding infinite tree-structured PGM.
The authors describe an algorithm to construct a tree-structured PGM corresponding to a given DNN by unrolling it into a tree and making L copies of each non-output node (corresponding to an L-sample approximation of the expected value at the node). As L tends to infinity, the constructed PGM becomes equivalent to the DNN.

A key implication of this result is that PGM algorithms can now be used to train DNNs. Specifically, the authors observe that the Stochastic Gradient Descent (SGD) approach to training DNNs differs from the contrastive divergence (CD) algorithm since SGD ignores the values of the output variables when computing the expected values of the latent variables. Using this observation, the authors propose a CD-based learning algorithm for DNNs that uses Hamiltonian Monte Carlo (HMC) sampling to approximate the expectations instead of Gibbs sampling which would be computationally inefficient given the large number of variables in the DNN.

The authors evaluate the proposed HMC-based learning algorithm using synthetic data sets and the Covertype data set from the UCI Machine Learning repository. They compare their approach against SGD and Gibbs sampling-based learning. They measure the calibration of the models learned on the synthetic data sets using Mean Absolute Error (MAE) from the true distribution and they measure the calibration for the Covertype data set using Expected Calibration Error (ECE). The evaluation shows that the proposed HMC-based algorithm (1) is equivalent to SGD for large values of L and (2) yields highly calibrated models for small values of L (such as 10)

**Strengths:**

1. The work establishes a formal connection between DNNs and (infinite tree-structured) PGMs. Apart from being a new perspective on DNNs, this allows access to PGM theory and algorithms for learning and inference.
2. The authors show that the DNN forward pass is not equivalent to exact probabilistic inference on the same structure but it is equivalent to exact inference on a separate infinite tree-structured PGM that can be constructed from the DNN. This allows us to characterize the approximation made by forward pass-based inference in DNNs.
3. The authors use the infinite tree-structured PGM equivalence result to develop an HMC-based learning algorithm for DNNs and empirically show that the algorithm yields more calibrated models.

**Weaknesses:**

While the paper makes important contributions, the presentation makes it hard to read in some places
1) Lines 127-143 would be clearer if presented in an algorithm environment with appropriate notation.
2) LineD- 245 says "our previous C1 algorithm" but CD-k and CD-1 are explained in lines 264-266
3) The description of a BN in lines 270-281 seems out of place and should probably be in Section 5.1
4) The Experimental Results section should be reorganized for readability by explicitly discussing research questions, metrics, baselines, and data sets before presenting results and answering research questions.

While the authors contrast their work against Bayesian Neural Networks, they should also contrast it with Probabilistic Circuits such as sum-product networks which are computational graphs that define a joint distribution over a set of input variables.

**Questions:**

Can you elaborate on the working of the Gibbs sampling baseline in the experimental evaluation? Does it also depend on L?
Also, how many samples were used for each of the experiments?

**Limitations:**

The authors clearly state the limitations of the work such as applicability to non-sigmoid DNNs and describe ways in which future work could address them.

---

> ### Author Rebuttal · Authors · 2024-08-07
>
> Thank you very much for your detailed review, great comments on the paper presentation and questions about the experiments. The first weakness about the PGM-DNN construction is addressed in the general response, and we also agree that some rearrangements in Section 5 would be helpful to make the presentation more clear as you mentioned in the other weaknesses. We will include an explanation to CD-1 and CD-k when we mentioned contrastive divergence in the first paragraph of Section 5, and move the description of a BN in lines 270-281 to the beginning of section 5.1 in the revision.
>
> For the fourth weakness about the result section, the discussions on research questions and baseline are mentioned at the beginning of Section 5.2, and the descriptions about datasets and metrics are at the beginning of Section 5.2.1 and 5.2.2 since we are using different types of datasets in our experiments. Thank you for pointing out that the results should be put after the discussion and we will modify the position setting of Table 1 and move it to the end of Section 5.2.1.
>
> For the first question about Gibbs sampling in the experimental evaluation, both Gibbs and HMC are sampling methods to generate new states for all the hidden nodes and they both apply CD-1 or CD-k to learn the weight. While HMC samples real values in the range of $[0, 1]$ under the normal distribution, Gibbs samples values in {0,1} under Bernoulli distribution. For node $h_{ij}$, i.e., the $j$th node in $h_i$ layer, the Bernoulli distribution used to sample its new state is determined by its Markov blanket, which includes all the nodes in layer $h_{i-1}$, $h_i$, and $h_{i+1}$ in this case. It could be calculated by normalizing the joint probability distribution of the blanket when $h_{ij}=1$ versus $h_{ij}=0$, which is written as following:
> $$
> P(h_{ij}=1)=\frac{P(h_{ij}=1|h_{i-1})\cdot P(h_{i+1}|h_i\setminus\\{h_{ij}\\},h_{ij}=1)}{P(h_{ij}=0|h_{i-1})\cdot P(h_{i+1}|h_i\setminus\\{h_{ij}\\},h_{ij}=0)+P(h_{ij}=1|h_{i-1})\cdot P(h_{i+1}|h_i\setminus\\{h_{ij}\\},h_{ij}=1)}
> $$
>
> We then sample a new state for $h_{ij}$ from $\text{Bern}(P(h_{ij}=1))$. In Gibbs sampling, the node is sampled one by one since a newly sampled node will affect the sampling of subsequent nodes in its Markov blanket. After sampling new values for all the hidden nodes, the weight is similarly updated using Equation (5)(6)(7) (line 302). So, the Gibbs sampling we used does not depend on $L$.
>
> For the second question about the number of samples, we use 1000 samples for each experiment, and train-test split ratio is 80:20. These are also explained at the beginning of Section 5.2.1, 5.2.2 and Appendix E.

---

> > ### Comment · Reviewer_7PML · 2024-08-13
> >
> > Thank you for the response. I understand the work more clearly now. After going through other reviews and the discussion, I have updated my rating to Accept

---

### Official Review · Reviewer_zi4w · 2024-07-20

**Soundness:** 2
**Presentation:** 2
**Contribution:** 3
**Rating:** 3
**Confidence:** 3

**Summary:**

This paper re-interprets deep neural networks (with sigmoid activations) as probabilistic models.  The authors give a construction that converts a DNN to an infinite tree-structured PGM, which has the benefit of yielding probabilstic information about its internal nodes.  The key step is to copy input nodes so that they are only used once, and that their expectation functions as their value. The authors then prove that, in a certain infinite limit, the semantics are the same as of the original DNN.  They then set up synthetic experiments using Hamiltonian Monte Carlo sampling on an approximation to this PGM, and report that their results help to reduce overfitting in small synthetic datasets.

**Strengths:**

The approach to understanding the internals of a DNN probabilistically are novel, and it seems like an approach worth pursuing at a high level.  The introduction is well written, modulo a few techincal gripes.  The biggest case to be made in favor of this construction (perhaps augmented with HMC) is that it might help reinterpret a trained network in probabilistic terms, avoiding pitfalls with maximum likelihood in "post-processing".  The authors set up experiments that seem to show this to a small degree in a synthetic setting.  The approach is creative and interesting.

**Weaknesses:**

The paper has major weaknesses in both the theoretical and the experimental parts.  On the theoretical side, the constructions  are imprecisely presented, and it is not at all clear to me that the main theoretical contribution of the paper (Theorem 1) delivers on the promises made on the introduction, nor that it is pedagogically useful.  At a lower level, it seems there are a lot of minor bugs and ambiguities and undefined symboles that have been swept under the rug.  At a minimum, I think the theoretical part of this paper needs a major rewrite with an eye towards precision and detail.

On the empirical side, I'm afraid the experiments are not totally convincing either.  My biggest problem is the lack of standard baselines. Why go all the way to HMC?  The setup seems a bit rigged: by training a DNN with SGD on such a small number of samples, you are bound to see overfitting, and the new methods get to start from the maximum likelihood solution found by SGD. Why is the "cold-start" version, which does not train with SGD at all not reported? What about regularized approaches, or ones in which the samples are not fixed? These numbers would really help to contextualize the preformance of the new system.  Overall, it seems that the authors have managed to "make the system work", but have not invested very much time in verifying that it works better than simple alternatives.

I appreciate that the authors added additional experiments in the appendix, but they are not flattering.  The HMC for the real world data works best for the smallest L (which seems antithetical to the main story showing that $L=\infty$ is appropriate), and in one case (=1/4), the new methods are significantly worse than the original DNN performance. In addition, the newly proposed methods are 2-3 orders of magnitude slower than the baseline.

-----
minor line-by-line comments:
-  I disagree that VAEs are a "link between graphical models and deep neural networks" (line 17). The "graphical models" part of VAEs is relatively tenuous; it's really just a way to emphasize a data generating process by which samples are drawn from a prior and then decoded.  To make the connection to PGMs you'll probably need a much more expressive variant, such as a PDG (https://arxiv.org/abs/2202.11862).
- The notation in lines 74-75 is a bit slopy; is $v_i$ a variable, or a setting of a variable? If the former, then the equation $p(\vec v) = $... doesn't typecheck; if the latter, then it shouldn't be the argument to $pa(v_i)$. I suggest just using $pa_i$ for a quick and dirty solution.
- I found the natural language description of steps b and c (lines 132-141) confusing, imprecise, and unnecessarily verbose. I do not feel I could implement it based on the description given, although Figure 1 helps. Pseudocode, or adding mathematical symbols to refer to nodes and edges, could make things far less ambiguous.
- Similarly, the discussion in lines 146-164 is also difficult to follow, and feels imprecise. I would like to see a more formal mathematical description of this.
- The usage of "mean-field approximation" on lines 148-149 is non-standard and seems like a  false connection to me.
- I disagree that this doubly exponentially large tree-structured graphical model is "easier to understand" than the original forward pass of the network (lines 181-184).  I also do not see why the training procedure of SGD is relevant to the discussion; how have you broken the symmetry between SGD and any other local optimization procedure? SGD certainly does not appear in Theorem 1.
- the notation $\sigma$ was not defined. Based on the appendix and standard symbols I gather it's the standard sigmoid curve, but this usage of $\sigma$ should be mentioned in the formal statement of the theorem or in the preliminaries.

**Questions:**

What happens if you don't do many epochs on the same fixed dataset, but rather draw fresh samples from the ground truth distribution that you're looking for?  It seems that the main benefit shown in the experiments is that it can mitigate overfitting.  But at a high level, it's unsurprising that a strong structural "prior" about the data, which happens to be correct in the synthetic setting, can help overfit to this setting. What happens if the number of datapoints grows far larger, but we reduce the number of epochs?

Why is Hamiltonian Monte Carlo a particularly natural fit for this setting?


LOW-LEVEL QUESTIONS AND COMMENTS:

(Lines 96-99): Is the interpretation of a DNN with sigmoid activations really a coherent interpretation?  Yes, it defines a distribution over a binary variable given values of its parents---but in order for that cpd to typecheck, the parents need to take on real values, not binary ones. It would seem that this makes the interpetation inconsistent for networks deeper than one layer. ~~Am I missing something?~~
Edit: this is a confusing presentation, and the confusion deepens in lines 112-114. But the resolution in lines 116-119 is clear, and that should have been present from the beginning.  There's no reason to give a high-level overview of an interpretation that is confusing (because it is technically wrong) before pointing out that it is technically wrong.  The authors define this to be "the direct PGM" far later in the paper (line 190),

In line 121, the autors say that this "approximation of $\cal D'$ to $\cal D$ is precise", but is it? What is the distribution $\cal D'$? Over what variables? Are they continuous or binary? It still seems inconsistent to me. I would like to see this spelled out precisely. If it is meant as only a rough analogy, then you must say so when you introduce it.

Clarification about the "second step" of the construction: it seems to me it must be done backwards, so the second-to-last layer (after the "first step") has $L$ copies, the one before that has $L^2$ copies, and the first one has $L^n$ copies.  Is this correct?  Or are there overall $L$ copies of each layer of the "post-first-step" transformation?

Why is the variance $p_{ij} (1-p_{ij})/L$ in equation 1?  Is this shown in the appendix, or do you have a reference?

Is there some relation between the $M^{-1}$ covariance and the weight $\mathbf W$? Where do you get the esimtate of $M^{-1}$?

Shouldn't $\boldsymbol \mu$ factor into the update in equation (7) somewhere?

How are the procedures "Weight-updating" and "Sampling" in Algorithm 1 defined? Does this have to do with equation 7? I feel some details are missing.

The results in lines 334-337 are compelling, but it seems the opposite is true of the MN rows of the table.  Why is this?

**Limitations:**

The authors do discuss limitations a reasonable extent.

---

> ### Author Rebuttal · Authors · 2024-08-07
>
> Thank you for your detailed and constructive review. We take the comments and questions in order, except that the first criticism of the theory is general and leaves the specifics for later, so we take those later, in the order they are given.
>
> - *"Lack of standard baselines"* The baselines are SGD and Gibbs; the first is exactly correct with respect to the DNN and the second is exactly correct with respect to a direct PGM intepretation of the DNN as a Bayesian belief network.  Since Gibbs is excessively slow in this setting, we begin with SGD and then refine with Gibbs for the Gibbs baseline.
>
> - *On comparison to regularized approaches or use of larger/unlimited datasets.* Early stopping is the most commonly used, successful regularizer in SGD, so in every case we compare 100 epochs of SGD against 1000 epochs.  In most cases the message is similar for both 100 and 1000 epochs, with the exception when we get to the largest weights in the ground truth Markov network; here regularization *hurts* rather than *helps* the baseline of SGD.
>
> - *"The HMC for the real world data works best for the smallest L..."* The updated results (attached) with repeated training runs now agree exactly with what the theory predicts, suggesting that previous inconsistencies were due to random variability. As L gets small, HMC gets close to Gibbs (Bayesian belief net as ground truth), and as L gets large, HMC gets close to the DNN forward pass (infinite-width tree-structured PGM as ground truth).  When the ground-truth DNN weights are large, this difference is great, and it is more critical to agree with the infinite-width tree-structured PGM; here HMC1000 wins, although it doesn't beat SGD provided SGD gets sufficient epochs (1000 rather than 100).  When the ground truth DNN weights are small, Gibbs and the faster HMC10 win.
>
> - *"I disagree that VAEs"*  VAEs are still commonly regarded as among the first works in the progression linking DNNs and PGMs, so we prefer to retain this citation, but we will certainly cite the suggested reference as an important step in this progression.
>
> - *"The notation in lines 74-75"* It is common to write $P(X_1,\cdots,X_n) = \prod_i P(X_i|Pa(X_i))$ to stand as holding over all the possible different settings of $X_1,\cdots,X_n$ in defining the joint distribution represented by a Bayesian network (see Russell and Norvig textbook for example). But we see how our use of lower-case variable names here might have confused matters, so we will use upper-case variables names and clarify our meaning beforehand, also with a citation.
>
> - *"mean-field approximation on lines 148-149"* The term mean field approximation was used in that section due how the forward pass of the neural network might be viewed as using the expectation of the nodes of the previous layer to calculate their their value, which aligns somewhat with the mean field approximation described in: Jun Zhang, "The Mean Field Theory In EM Procedures For Markov Random Fields," \textit{Proceedings of the Seventh Workshop on Multidimensional Signal Processing}, Lake Placid, NY, USA, 1991, pp. 9.1-9.1, doi: 10.1109/MDSP.1991.639423. That said, the connection is indeed underexplored in this paper and differs notably in how the neural network only ever passes information forward. A more precise description, that the neural network effectively takes a local average from the immediately previous nodes and passes that information only forward, is more apt and we will revise.
>
> - *"I disagree that this doubly exponentially large tree-structured graphical model is `easier to understand'..."*  We meant specifically that the tree-structured graphical model represents a full joint probability distribution over all the nodes in the neural network under the standard semantics of all Markov networks. We will tighten the language to clarify.
>
> - *"I also do not see why ... SGD is relevant to the discussion"* Our MN is well-defined regardless of SGD. But the point of the theorem is that for every node, the probability it is true in the MN equals its output in the DNN.  As a result, the MN interpretation gives the same gradient that SGD uses.  So SGD training is correct with respect to this MN, and this MN is correct with respect to SGD training.  We wanted the MN to not be just some representation of the DNN, but the joint probability distribution with respect to which SGD in the DNN is correct.
>
> - *"(Lines 96-99)"* Thank you for the catch, the DNN indeed does not align with Bayesian networks or Markov networks outside of the case where a single hidden node is fully surrounded by observed evidence. We will clarify in this section how the DNN might be viewed as an approximation and that the probabilities calculated otherwise would not align.
>
> - *"In line 121, ... this 'approximation of $\cal D'$ to $\cal D$' is precise"* We stated that the approximation of $\cal D'$ to $\cal D$ is precise because, given a weighted acyclic graph with binary input variables, the neural network's forward pass can be used to calculate an 'approximate' conditional probability for any unobserved node. The values provided by that forward pass would not match the exact conditional probabilities of a similarly structured Bayesian network, but even without the PGM construction, they are well defined and could be used as a poor approximation.
>
> - *"Why is the variance $p_{ij} (1-p_{ij})/L$ in equation 1?"* This follows from the normal approximation to the binomial and the fact that $\text{Var}(X/L) = \text{Var}(X)/L^2$ for constant $L$. We will clarify.
>
> - *Details of Section 5.1.* In the final version, we will clarify the relationship between $M^{-1}$ and $\mathbf{W}$ and role of $\mu$ in equation (7).
>
> - *"How are the procedures 'Weight-updating' and 'Sampling' in Algorithm 1 defined?"* Equation 7 does indeed define the weight update step of the proposed algorithm. The sampling step is done using an MCMC chain (specifically HMC) as described in section 5.1 and Appendix C.

---

> > ### Comment · Reviewer_zi4w · 2024-08-13
> >
> > Thank you for answering my questions, and running additional experiment. This rebuttal has allayed some of my concerns, and I now am more on board with the empirical evaluation.
> >
> > Still, I'm not fully convinced.  A few responses:
> >
> >  - SGD: I was trying to point out that SGD is an optimization procedure. The "stochastic" part is just to speed things up computationally; the theory seems to me only to interact with the "gradient descent" part--- and even then, I believe you are referring to gradient descent with respect to a specific loss function (cross entropy, presumably).  I'm not sure what it means to say *"the MN interpretation gives the same gradient that SGD uses."*  Why does the MN have a gradient, and with respect to what? I'm lost.
> >
> >  - line 121:  you still have not given a coherent joint distribution $\mathcal D'$. Just because the DNN gives "approximate" answers to certain queries in some sense does not mean there is a joint distribution $\mathcal D'$ that coincides with those answers.
> >
> > - If you are going to have "weight updating" and "sampling" in your algorithm pseudocode, you should make sure to define those terms eslewhere.  Alternatively, write "update weights according to (7)" or something like this, so that a reader who is less intimate with the material can follow the references.

---

> ### Author Response · Authors · 2024-08-13
>
> (1) 'SGD':
>
> You are right our result applies to gradient descent.  We will clarify this.  The stochastic part of SGD comes entirely from which subset of examples (mini-batch) we use to compute this gradient at any step, so it applies to SGD as well.  We will clarify that the point is about gradient descent generally, not SGD in particular.
>
> Yes, MN learning is also by gradient descent.  And yes, our result only applies to cross entropy error, since cross entropy is a function of $P(y|X)$ for any setting of the inputs $X$.  Our result is that these probabilities $P(y|X)$ agree between NN and MN for any node y anywhere in the NN, not just output nodes.  We will make sure this limitation to cross entropy error is extremely clear; thank you for bringing to our attention the need for more clarity and emphasis on that point.
>
> (2) 'coherent joint distribution $\mathcal D'$':
>
> Thank you for bringing attention to this detail. You are correct that despite it being possible for the neural network's standard operation to be viewed as an 'approximate' marginal $p(y|X)$ for the output or $p(h|X)$ for a hidden node, that view (and less exact connections with probabilistic graphical models) does not immediately lend itself to a distribution over multiple nodes (or the full joint distribution), $p(y, h_1, h_2, h_3 |X)$ for example. In the traditional neural network structure, those hidden neurons are not well-defined unobserved nodes of a PGM and we therefore require the infinite PGM view. The infinite width tree structured PGM view should then be able to connect the neural network's forward pass and these 'approximate' values with an exact probabilistic structure where queries on multiple nodes, jointly, is possible. We will make sure to clarify that point in the final version of the paper.
>
> (3) '"weight updating" and "sampling" in your algorithm pseudocode':
>
> Thank you for catching that detail. We will make sure to clarify the weight updating and sampling steps of Algorithm 1 in the final version of the paper.

---

### Author Rebuttal · Authors · 2024-08-07

We thank all reviewers for the thoughtful and thought-provoking reviews. Here we list concerns raised by multiple reviewers and describe our plans to address them. *If a change recommended by reviewers is not explicitly addressed, this implies that we will follow the reviewer's recommendation.*

**1. Clarity of PGM Construction.** Several reviewers raised concerns about the clarity of the plain language description of the construction of the probabilistic graphical model whose conditional probability exactly matches the forward pass of the neural network. We acknowledge the need to complement these descriptions with a more precise presentation. To do this, we now use the algorithm environment to precisely define (a) Step 1 of our construction, in which we unroll the initial, likely loopy, neural network graph into a tree structure [Algorithm 1], and (b) Step 2 of our construction, in which we create infinite copies of each node and subtree necessary to exactly match the forward pass [Algorithm 2]. These two Algorithms may be found in the 1-page attachment to our response. We also intend to refine the existing plain language descriptions in the final version. By combining these two approaches, our hope is to provide a mathematically precise, reproducible description while still retaining the intuition that we believe the plain language description provides.

**2. Strength of Experimental Results.** Several reviewers also noted that they would like to see more thorough experimental results for the proposed HMC algorithm. We agree that this is key, and we have been working to provide a more comprehensive set of results that include repeated training runs and thereby allow us to determine whether differences between each training method and the baseline approach (DNN trained via SGD) are statistically significant. These results now clearly illustrate settings in which HMC-10 is favorable compared to the DNN. Specifically, HMC-10 performs best in settings where the relationship between inputs and outputs tends to be more random (lower valued weights). The results also show that HMC-10 succeeds in some of the settings where Gibbs fails. Finally, results underscore the similarity between DNN performance and HMC-100/1000 performance, suggesting that HMC-L more closely approximates DNN training as L increases, as predicted by our theoretical results.

**3. Real-World Applications.** While we agree that highlighting real-world applications will be important as we work to establish the practical benefits of this new connection between PGMs and DNNs, the current work is primarily theoretical in nature, with complementary experimental results that (a) corroborate the theory, and (b) illustrate a possible direction suggested by this new connection. Moreoever, from a practical perspective, we require much larger sample sizes to demonstrate benefit in a real-world binary classification setting, because in such settings -- in contrast to our simulations -- the true event probabilities are not known. In future work, we do plan to explore benefits of our HMC approach in real-world datasets, and we also hope to establish other practical, applications-oriented benefits of this new PGM view of DNNs.

---

> ### Author Response · Authors · 2024-08-12
>
> Thanks again to all reviewers for your constructive comments. We believe the additional empirical results (see attached pdf) serve to strengthen our claims, and that other changes we've outlined will substantially improve the presentation of this work. If any reviewers have follow-up questions or comments, please let us know.

---

### Decision · Program_Chairs · 2024-09-25

**Decision:**

Accept (poster)

**Comment:**

This paper shows how to re-interpret deep neural networks with sigmoid activations as Bayesian networks.  The reviewers have rather mixed opinions. One reviewer points out some (potential) flaws in the DNN interpretation in terms of discrete versus continuous RVs. The rebuttal clarified quite some of the issues raised. However, the DNN interepretation seems not to be highgly novel as it is very much related to learning message-passing inference, as e.g. considered by Ross et al. (Learning message-passing inference machines for structured prediction, CVPR 2011); this should be clarified. INdeed, this is starting from a BN, but it also suggest that there is a tight connection. Anyhow, the paper is not just about the DNN interpretation but also about the new sampling approach derived via the interpretation. This also addresses to some extend another point raised by one reviewer, namely that the "practical utility of going from the DNN function space to the PGM space is yet unknown"; the development of new algorithm is definitely something interesting. So overall, reducing the weight of the negatgive review (as the rebuttal addressed many of the points), the connection between PGMs and DNNs is beneficial and worth exploring and revisiting (including connections to some of the older works that other reviewers have mentioned).